# From simple labels to semantic image segmentation: Leveraging citizen science plant photographs for tree species mapping in drone imagery

Salim Soltani[1,2,3*], Olga Ferlian[3,5], Nico Eisenhauer[3,5], Hannes Feilhauer[1,2,3,4], and Teja Kattenborn[1,3]

[1]Remote Sensing Centre for Earth System Research (RSC4Earth), Leipzig University, Germany
[2]Center for scalable data analytics and artificial intelligence (ScaDS.AI), Leipzig University, Germany
[3]German Centre for Integrative Biodiversity Research (iDiv), Halle-Jena-Leipzig, Germany
[4]Helmholtz Centre for Environmental Research, Leipzig, Germany
[5]Institute of Biology, Leipzig University, Germany
[*] Corresponding author: salim.soltani@uni-leipzig.de

**Abstract**

Knowledge of plant species distributions is essential for various applications, such as nature conservation, agriculture, and forestry. Remote sensing data, especially high-resolution orthoimages from Unoccupied Aerial Vehicles (UAVs), were demonstrated to be an effective data source for plant species mapping. Particularly, in concert with novel pattern recognition methods, such as Convolutional Neural Networks (CNNs), plant species can be accurately segmented in such high-resolution UAV images. Training such pattern recognition models for species segmentation that are transferable across various landscapes and remote sensing data characteristics often requires excessive training data. Training data are usually derived in the form of segmentation masks from field surveys or visual interpretation of the target species in remote sensing images. Still, both methods are laborious and constrain the training of transferable pattern recognition models. Alternatively, pattern recognition models could be trained on the open knowledge of how plants look as available from smartphone-based species identification apps, that is, millions of citizen science-based smartphone photographs and the corresponding species label. However, these pairs of citizen science-based photographs and simple species labels (one label for the entire image) cannot be used directly for training state-of-the-art segmentation models used for UAV image analysis, which require per-pixel labels for training (also called masks). Here, we overcome the limitation of simple labels of citizen science plant observations with a two-step approach: In the first step, we train CNN-based image classification models using the simple labels and apply them in a moving-window approach over UAV orthoimagery to create segmentation masks. In the second phase, these segmentation masks are used to train state-of-the-art CNN-based image segmentation models with an encoder-decoder structure. We tested the approach on UAV orthoimages acquired in summer and autumn on a test site comprising ten temperate deciduous tree species in varying mixtures. Several tree species could be mapped with surprising accuracy (mean

F1-score = 0.47). In homogenous species assemblages, the accuracy increased considerably (mean F1-score 0.55). The results indicate that many tree species can be mapped without generating training data and by integrating pre-existing knowledge from citizen science. Moreover, our analysis revealed that citizen science photographs' variability in acquisition data and context facilitates the generation of models that are transferable through the vegetation season. Thus, citizen science data may greatly advance our capacity to monitor hundreds of plant species and, thus, Earth´s biodiversity across space and time.

Keywords: Remote Sensing, Convolutional Neural Network, Citizen Science Data, Plant species, Transfer learning.

## 1 Introduction

Spatially explicit information on plant species is crucial for various applications, including nature conservation, agriculture, and forestry. Remote sensing emerged as a promising tool to create spatially continuous maps of plant species (Müllerová et al., 2023; Bouguettaya et al., 2022; Fassnacht et al., 2016). Thereby, supervised machine learning algorithms are commonly used to identify species-specific features in spatial, temporal, or spectral patterns of remotely sensed signals (Sun et al., 2021; Maes and Steppe, 2019; Lopatin et al., 2019; Curnick et al., 2021; Wagner, 2021). In recent years, remote sensing imagery from drones, also known as Unoccupied Air Vehicles (UAVs), has emerged as an effective source of information for mapping plant species (Kattenborn et al., 2021; Fassnacht et al., 2016; Schiefer et al., 2020). By means of mosaicing a series of individual image frames, UAVs enable the creation of georeferenced orthoimagery of relatively large areas with extremely high spatial resolution, e.g., in the mili- or centimeter range. The fine spatial grain of such imagery can reveal distinctive morphological plant features to identify specific plant species. Such plant features include the leaf shape, flowers, branching patterns, or crown structures (Sun et al., 2021; Kattenborn et al., 2019a). An effective way to unleash the potential of these fine spatial features is given by deep learning-based pattern-recognition techniques, in particular by Convolutional Neural Networks (CNN). A series of studies have demonstrated that CNN can precisely segment plant species' crowns in high-resolution UAV imagery (Kattenborn et al., 2021; Hoeser and Kuenzer, 2020; Brodrick et al., 2019). Such CNN models learn the characteristic spatial features of the target (here, plant species) through a cascade of filter operations (convolutions). Given these high-dimensional computations, efficiently adopting these models to UAV orthoimagery, with their large spatial extents but also high resolution, requires training and applying them sequentially using smaller sub-regions of an orthoimage (e.g., image tiles of 512 by 512 pixels, Fig. 1a).

However, generating models that are transferable across various landscapes and remote sensing data characteristics requires large amounts of training data (Kattenborn et al., 2021; Galuszynski et al., 2022). In particular, when neighboring plant species bear a similar resemblance, a wealth of training data becomes essential, allowing the model to discern the subtle distinctions between these species (Kattenborn et al., 2021; Schiefer et al., 2020). Commonly, the generation of training data is costly. Training data are usually derived from field surveys

or visual interpretation of remote sensing images, also known as annotation or labelling. Both methods have limitations: Field surveys are often logistically challenged by site accessibility or travel costs. Moreover, field surveys commonly only enable the acquisition of point observations or relative cover fractions of the target species (Leitão et al., 2018). Visual image interpretation is often much more effective (Kattenborn et al., 2019b; Schiefer et al., 2023) but for some species, precise visual identification of species can be challenging due to subtle indicative morphological features, the variability of these features in the landscape, or the complexity of vegetation communities (e.g., smooth transitions of canopies of different species). Moreover, the representativeness of data derived from field surveys and visual interpretation is often limited to the location where and when the data were acquired, which may reduce a model´s generalization to new regions or time periods (Kattenborn et al., 2022). Therefore, the obtained amount and quality of training data can be a critical factor for the performance and transferability of CNN models (Bayraktar et al., 2020; Rzanny et al., 2019; Brandt et al., 2020).

The challenge of limited training data for UAV-based plant species identification may be alleviated by the collective power of scientists and citizens openly sharing their plant observations on the web (Ivanova and Shashkov, 2021; Fraisl et al., 2022; Di Cecco et al., 2021). A particular data treasure in this regard is generated by citizen science projects for plant species identification. Examples are the iNaturalist and Pl@ntNet projects, which encourage ten-thousands of individuals to capture, share, and annotate photographs of the World´s plant life (Boone and Basille, 2019; Di Cecco et al., 2021). The quantity of such citizen science observations is rapidly growing due to the increasing number of volunteers participating in the platform (Boone and Basille, 2019; Di Cecco et al., 2021).

Currently, the iNaturalist project contains over 26 Mio of globally distributed and annotated photographs of vascular plant species. The iNaturalist platform allows users to identify plant species manually or using a computer vision model integrated into the platform. The submitted observations are then evaluated by the community, and a research-grade classification is assigned if over two-thirds of the community agrees on the species identification. The Pl@ntNet project includes over 20 Mio observations of globally distributed vascular plants. Pl@ntNet requires users to photograph their observations and select an organ tag (e.g., leaf, flower, fruit, or stem). The Pl@ntNet features an image recognition algorithm to analyze the tagged photograph and suggest a plant species. Pl@ntNet's validation process uses a dynamic approach, combining automated algorithm confidence with community consensus (Joly et al., 2016). The validated observations of iNaturalist and Pl@ntNet are shared via the Global Biodiversity Information Facility (GBIF), a global network that provides open access to biodiversity data (GBIF, 2019).

Citizen science-based plant photographs with species annotations provide a valuable, large, and continuously growing data source for training pattern recognition models, such as CNNs (Van Horn et al., 2018; Joly et al., 2016). However, such citizen science data has a cardinal limitation: It only provides simple species annotation for a plant photograph (*the image$_i$ shows species$_j$*). Hence, these labels only enable to train image classification models that

predict the likelihood of a species being present in an image but not where in the image. In an ideal setting for species mapping applications, the species labels would delineate the regions or pixels belonging to a species (*The pixels in the right corner of image*$_i$ *represents species*$_j$). Such labels (known as masks) could be used to train CNN-based segmentation models, which can predict a species probability for each individual pixel of an image (or tile of an orthoimage) (Galuszynski et al., 2022; Schiefer et al., 2020).

In a pioneering study by Soltani et al. (2022), the limitation of the simple labels that come with citizen science photographs was overcome by a workaround. At first, image classification models were trained with citizen science data and simple labels to predict a species per image. The trained image classification models were then applied sequentially on tiles of 512 × 512 pixels of UAV-based orthomosaics in a moving-window-like fashion with very high overlap (Fig. 1a). Lastly, the individual predictions derived from the moving-window steps were rasterized to a seamless segmentation map (Fig. 1b). However, this workaround is computationally intense and inefficient for large or multiple UAV orthomosaics, as segmentation maps can only be derived from many overlapping prediction steps. In contrast, state-of-the-art CNN-based segmentation methods (typically an encoder-decoder structure) used in remote sensing applications are trained with reference data in the form of masks with dimensions (pixels) corresponding to the extent of the imagery, where each pixel of the mask defines the absence or presence of a class (here plant species) in the imagery (Kattenborn et al., 2021). Respective segmentation models are more efficient as they segment multiple classes in a single prediction step. Moreover, they enable more detailed class representations in situations where multiple classes are arranged in complex patterns.

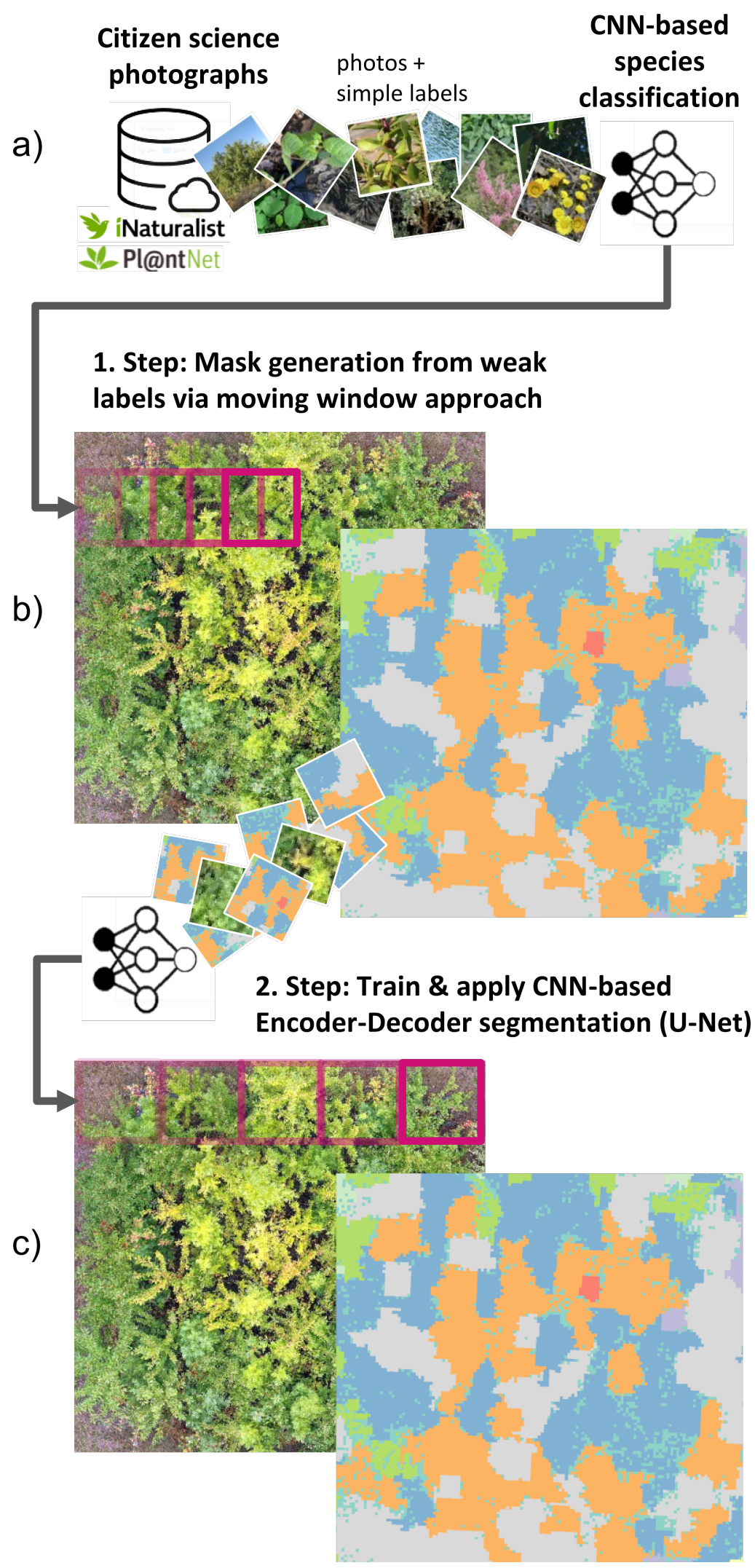

Figure 1: 1-column figure: Schematic representation of the proposed workflow, including the moving window approach by Soltani et al. (2022) (a,b) and the use of state-of-the-art encoder-decoder segmentation algorithms (c).

Here, we propose a solution to overcome the limitation of simple annotations of citizen science plant observations with a two-step approach: In the first step, we apply the procedure of Soltani et al. (2022), involving CNN-based image classification models trained on citizen science photographs and simple species labels to predict plant species in UAV orthoimages using the moving-window approach described above (Fig. 1a, b). Although computationally demanding, this serves to create segmentation masks for UAV orthoimages. In the second step, these segmentation masks are used to train more efficient CNN-based image segmentation models with an encoder-decoder structure (Fig. 1c). These more efficient models could then be applied to larger spatial extents or due new UAV orthomosaics (e.g. of different sites or

time steps).

The present study, hence, addresses the following research questions:

- Can we harness weak labels from citizen science plant observations to train efficient state-of-the-art semantic segmentation models?
- Do those segmentation models also increase the accuracy compared to the simple moving window approach?

These questions are evaluated on a tree species dataset acquired on an experimental site (MyDiv experiment, Bad Lauchstädt, Germany), where ten temperate deciduous tree species were planted in stratified and complex mixtures. The selection of this location is attributed to its harmonious coexistence of various plant species within a compact area.

# 2 Methods

## 2.1 Data acquisition and pre-processing

### 2.1.1 Study site and drone data acquisition

The MyDiv experimental site is located in Bad Lauchstädt, Saxony-Anhalt, Germany (latitude, 51°23' N, longitude, 11°53' E). The site comprises 80 plots composed in different configurations of ten deciduous tree species, including *Acer pseudoplatanus*, *Aesculus hippocastanum*, *Betula pendula*, *Carpinus betulus*, *Fagus sylvatica*, *Fraxinus excelsior*, *Prunus avium*, *Quercus petraea*, *Sorbus aucuparia*, and *Tilia platyphyllos* (Ferlian et al., 2018). Each plot measures 12 m by 12 m and contains 140 trees planted at distances of 1 m (Fig 2). In total, all plots together accommodate 11,200 individual trees. Each plot contains varying tree species compositions, including one, two, and four tree species. This variety in species, their balanced composition, and plots of different canopy complexity (species mixtures) provide an ideal setting to test the proposed species segmentation approach.

We collected UAV-based RGB aerial imagery over the MyDiv experimental site using a DJI Mavic 2 Pro and the flight planning software DroneDeploy (DroneDeploy vers. 5.0, USA). Two flights were conducted in 2022 in July and September, where July corresponds to the peak of the growing season and September to senescence stage (Fig 2). The flight plan was setup with a forward overlap of 90%, side overlap of 70% at an altitude of 16 m (ground sampling distance of approximately 0.22 cm per pixel). We used the generated images and Metashape (vers. 1.7.6, Agisoft LLC) to generate orthoimages for both flight campaigns. The orthoimage for July and September are onward called $\text{Ortho}_{\text{July}}$ and $\text{Ortho}_{\text{September}}$, respectively.

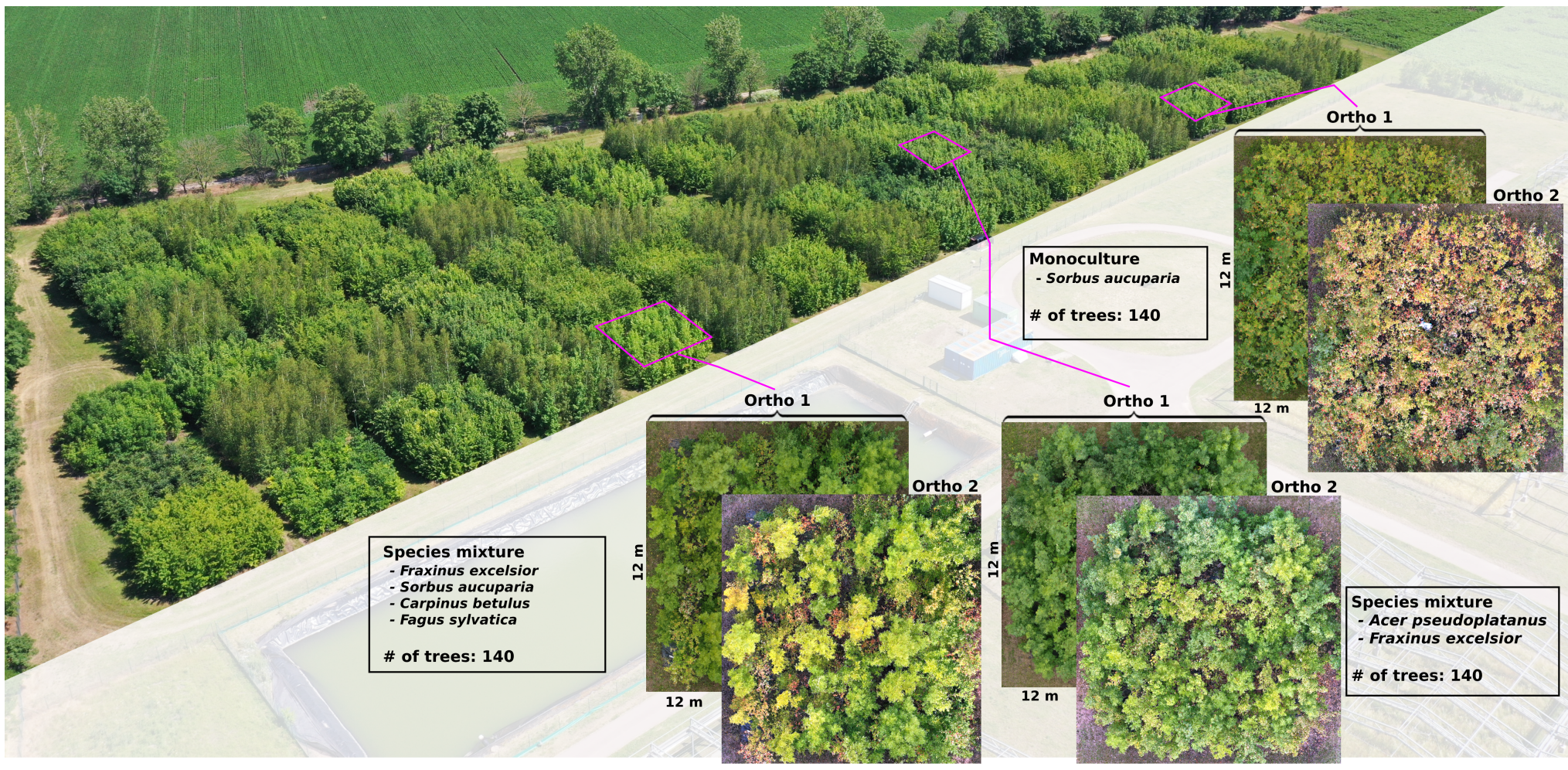

Figure 2: Overview of the MyDiv experimental site with close-ups for three plots of different species composition. The MyDiv site is located at Lat. 51.3916 N, Long. 11.8857 E.

To evaluate the performance of the CNN models for tree species mapping, we created reference data by manually delineating the tree species in the UAV orthoimages in QGIS (vers. 3.32.3). To reduce the workload, we did not delineate the species for the entire plot but for diagonal transects with 20 m length and 2 m width.

### 2.1.2 Citizen science training data

We queried plant observations of the iNaturalist and Pl@ntNet projects via the GBIF database for our target tree species using scientific names. For the iNaturalist data, we used the R package rinat (vers. 0.1.8), an API to iNaturalist. The Pl@ntNet data were acquired by submitting a download request for the selected tree species via GBIF. The number of photographs available from iNaturalist and Pl@ntNet varied for the different tree species. Per species, we were able to acquire between 582 to 10000 photographs (mean 7696) from the iNaturalist platform and 221 to 3304 images (mean 2238) from the Pl@ntNet platform (details see Appendix Table A1).

In addition to the tree species, we added a background class to consider canopy gaps between trees. For training data, we used the Google Image API to query different keywords, e.g.*grass*, *forest floor*, *forest ground*. After cleaning the obtained images for non-meaningful results, the background class included 1100 photographs.

We converted all photographs to a rectangular shape by cropping them to the shorter side and resampled them to a common size of $512 \times 512$ pixels (the tile size used later for the CNN model generation). Figure 3 shows examples of the downloaded photographs for the different tree species and a comparison to their appearance in $\text{Ortho}_{\text{July}}$.

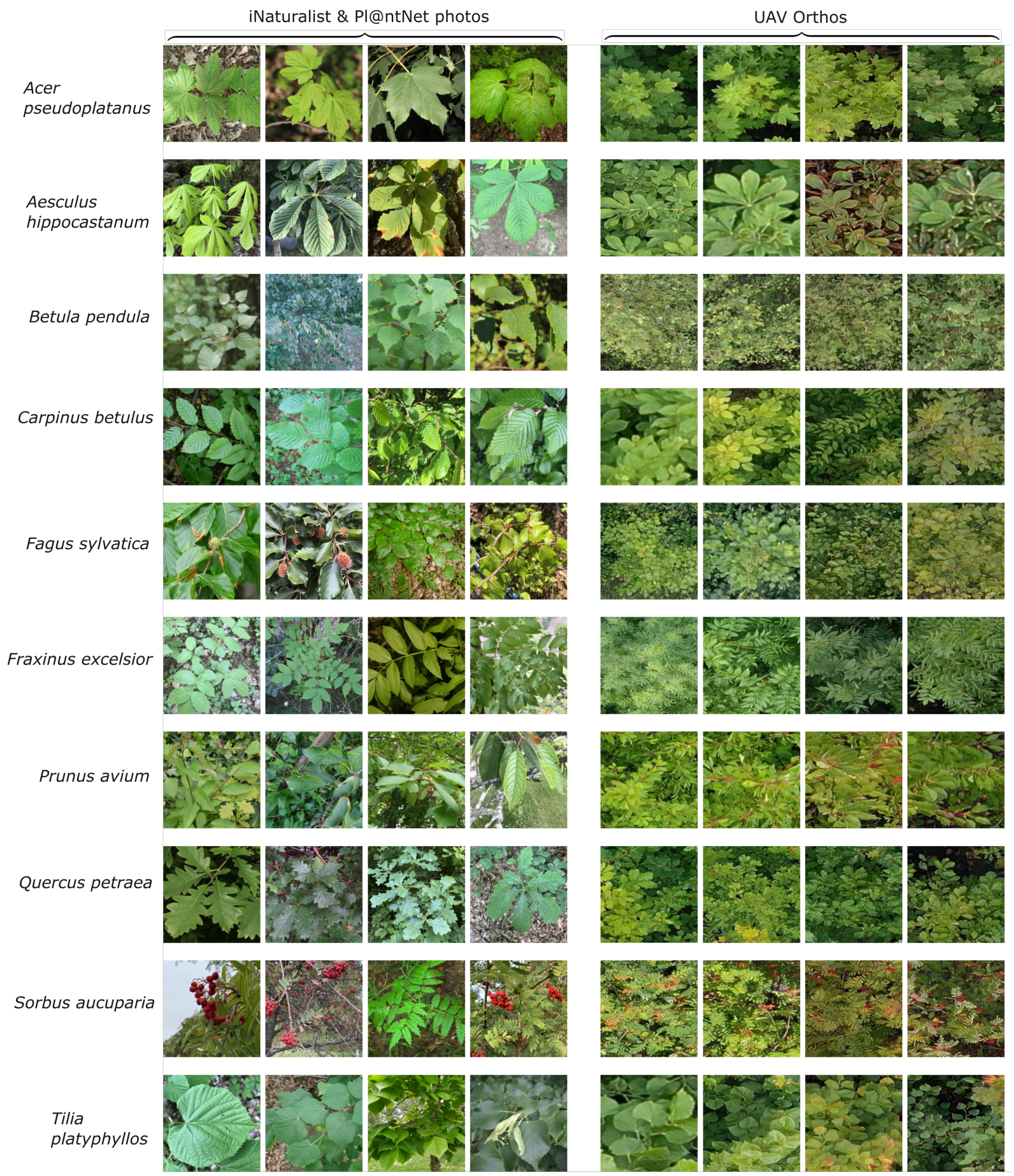

Figure 3: Example citizen science-based photographs derived from iNaturalist and tiles of UAV orthoimages (512 * 512 pixels) for the ten tree species in the MyDiv experiment.

The acquisition settings of citizen science plant photographs are heterogeneous and differ considerably from the typical bird perspective of UAV orthoimages. For instance, from the UAV perspective, canopies are mostly viewed from a relatively homogeneous distance, and the photographs represent mostly leaves and other crown components. In contrast, the citizen science data includes a lot of close-ups, landscape imagery, or horizontal photographs of

trunks. Soltani et al. (2022) has demonstrated that species recognition in UAV images can be improved by excluding crowd-sourced photographs that are exceptionally close (e.g., showing individual leaf veins) or too far away from the plant (e.g., landscape images). Accordingly, we filtered the citizen science-based training photos according to the camera-plant-distance. Moreover, we filtered photos that exclusively contained tree stems. Because such information is unavailable in the citizen science datasets, we trained CNN-based regression and classification models to predict acquisition distance and tree trunk presence for each downloaded photograph. To train these CNN-based models, we visually estimated the acquisition distance (4,500 photographs) and labeled tree trunk presence (1,000 photographs). To ease the labeling process, we used previously labeled training data from (Soltani et al., 2022) and added 150 additional tree photographs from the tree species present in the MyDiv experimental site.

To predict acquisition distance and trunk presence, We randomly split the citizen science-based plant photographs into training and validation sets, with 80% for training and 20% for validation.

For the distance regression and the trunk classification, we used the EfficientNetB7 backbone (Tan and Le, 2019). For the distance regression, we used the following top-layer settings: global average pooling, batch normalization, drop out (rate 0.1), and a final dense layer with 1 unit and linear activation function. We used the Adam optimizer (learning rate of 0.0001) and a mean squared error (MSE) loss function. For the trunk classification, we used the following top-layer settings: global max-pooling, a final dense layer with two units, and a softmax activation function. We used the Adam optimizer (learning rate of 0.0001) and the categorical cross-entropy loss function. Both models were trained using a batch size of 20 and 50 epochs.

We used the model with the lowest loss from these epochs (details on the model performance are given in Appendix A1.3) to predict the acquisition distance and tree trunk presence in all downloaded photographs for our target species. We filtered training photographs prior to training CNN-based species classification (see section 2.2) with acquisition distances less than 0.2 m and greater than 15 m and photographs classified as trunk (probability threshold of 0.5). Thereby, 82,628 of the 101,574 downloaded citizen science photographs remained.

## 2.2 CNN-based creation of plant species segmentation masks using a moving window approach

The segmentation masks were obtained using a CNN image classification model trained on crowd-sourced plant photographs and simple species labels using a moving window method (hereafter $\text{CNN}_{\text{window}}$, Fig. 1). Based on the results of previous studies, we choose a generic image size of $512 \times 512$ pixels for the CNN classification model (Schiefer et al., 2020; Soltani et al., 2022). During the moving window approach, the orthoimage is sequentially cropped into tiles of $512 \times 512$ pixels on which the image classification is applied to predict the species for each location. This procedure is applied with a dense overlap between tiles defined by a step size, resulting in a dense regular grid of species predictions. We chose a vertical and horizontal distance of 51 pixels as step size. The resulting predictions are afterward rasterized

to a continuous species distribution grid with a spatial resolution of 8.31 cm/pixel (see Soltani et al., 2022, for details). The $\text{CNN}_{\text{window}}$ model was implemented as a classification task with eleven classes, including the ten tree species and the background class.

The number of available photographs varied widely across tree species (see 2.1.2), potentially biasing the model towards classes with more photographs. To address this imbalance, we equally sampled 4,000 photographs for each class with replacement. We applied a data augmentation to increase the variance of the duplicated images. The augmentation consisted of random vertical and horizontal flips, random brightness maximum delta of 10% ($\pm 0.1$), and contrast alteration within a range of 90% to 110% (0.9 to 1.1) of training photographs. We randomly partitioned the training data into validation and training sets to ensure unbiased evaluation. We allocated a holdout of 20% of the training data for model selection, while the remaining 80% was used for model training. Subsequently, we assessed the accuracy of the selected model using independent reference data.

After testing different architectures as model backbones, including ResNet-50V2, EfficientNetB07, and EfficientNetV2L, we selected EfficientNetV2L. The following layers were added on top of the EfficientNetV2L backbone: Dropout with a ratio of 0.5, average pooling, dropout with a ratio of 0.5, dense layer with 128 units, L2 kernel regularizer (0.001), a ReLu activation function, and a final dense layer with a softmax activation function and 11 units. We used Root Mean Squared Propagation (RMSprop) as the optimizer with a learning rate of 0.0001 and categorical cross-entropy as a loss function. We trained the configured model with a batch size of 15 over 150 epochs. The model with the lowest validation loss (based on the 20% holdout) was selected as the final model. The latter was used to predict the tree species (probabilities) in the UAV orthoimages using the abovementioned $\text{CNN}_{\text{window}}$ method. To filter uncertain predictions (predominantly in canopy gaps or at crown shadows), we only considered a tree species as predicted above a threshold higher than 0.6. Otherwise, it was assigned to NA (not available). To smooth the predictions and remove noise, we applied a sieve operation on the output of the $\text{CNN}_{\text{window}}$ (threshold = 50, considering horizontal, vertical, and diagonal neighbors, R-package *terra*, vers. 1.7).

### 2.3 CNN-based plant species segmentation using an encoder-decoder architecture

As encoder-decoder segmentation architecture (onwards $\text{CNN}_{\text{segment}}$), we chose U-Net (Ronneberger et al., 2015), which is the most widely applied segmentation method in remote sensing image segmentation (Kattenborn et al., 2021). The U-Net architecture is a CNN-based algorithm that performs semantic segmentation by predicting a class for each pixel of the input image. The architecture consists of an encoder-decoder structure with skip connections. The configured architecture has four levels of convolutional blocks. Each convolutional block consists of two convolutional layers and is followed by batch normalization and ReLU activation. The encoder gradually compresses feature maps and reduces their spatial dimensions via max pooling operations, while the decoder increases the feature map resolution by transposed convolution. The encoder and decoder blocks are connected through skip connections, which

transfer the spatial context of the encoder feature maps to the decoder, enabling a segmentation at high-resolution in the last layer. The final layer has eleven units (corresponding to the ten tree species and a background class). A corresponding softmax activation function maps the features to class probabilities. Using a max function, the pixels of the segmentation output are assigned to the class with the highest probability (Fig. A12).

The segmentation masks for training $\mathrm{CNN}_{\mathrm{segment}}$ were obtained from the predictions of the $\mathrm{CNN}_{\mathrm{window}}$ method applied on both UAV orthoimages (section 2.2, $\mathrm{Ortho}_{\mathrm{July}}$, $\mathrm{Ortho}_{\mathrm{September}}$). At first, we resampled the $\mathrm{CNN}_{\mathrm{window}}$ prediction maps to the original spatial resolution of the orthoimages (0.22 cm pixel size). Afterward, we cropped the orthoimages and the prediction maps into non-overlapping tiles, each with a size of $512 \times 512$ pixels, resulting in a total of 44,980 and 37,113 tiles from $\mathrm{Ortho}_{\mathrm{July}}$ and $\mathrm{Ortho}_{\mathrm{September}}$, respectively.

The training data obtained from the $\mathrm{CNN}_{\mathrm{window}}$ approach were filtered to avoid training the $\mathrm{CNN}_{\mathrm{segment}}$ with uncertain predictions. Thereby, we assumed that higher model uncertainty are present in areas where the model predicts multiple classes with low relative cover. Thus, after initial tests, we included only those tiles where the cover of at least one class exceeded 30%. The number of training tiles per class after filtering varied between 1257 and 16894 samples; *Acer pseudoplatanus* (6581), *Aesculus hippocastanum* (2054), *Betula pendula* (4955), *Carpinus betulus* (1535), *Fagus sylvatica* (16894), *Fraxinus excelsior* (7901), *Prunus avium* (1257), *Quercus petraea* (1302), *Sorbus aucuparia* (5473), *Tilia platyphyllos* (1982), Background (5408).

Similar to the previous $\mathrm{CNN}_{\mathrm{window}}$ classification task, the availability of training tiles varied greatly across the tree species. This class imbalance may have partially stemmed from the more systematic misclassification of certain classes during the $\mathrm{CNN}_{\mathrm{window}}$ prediction. To reduce the unfavorable effects of a class imbalance on model training, we sampled 4,000 tiles per class with replacements (similar to the $\mathrm{CNN}_{\mathrm{window}}$ procedure). We applied the same data augmentation strategy as $\mathrm{CNN}_{\mathrm{window}}$ to increase variance among duplicates. 20% of the training data were withheld for model selection.

We trained the U-Net architecture using Root Mean Squared Propagation (RMSprop) as the optimizer with a learning rate of 0.0001 and an adapted Dice loss function. We adapted the Dice loss to ignore the weights coming from pixels with NA mask values. The models were trained with a batch size of 20 over 150 epochs.

The $\mathrm{CNN}_{\mathrm{segment}}$ was then applied to $\mathrm{Ortho}_{\mathrm{July}}$ and $\mathrm{Ortho}_{\mathrm{September}}$. To reduce uncertain predictions of $\mathrm{CNN}_{\mathrm{segment}}$, we assigned the pixels where predicted probabilities did not exceed 0.3 to the background class. Thereby, we assumed that uncertain predictions predominantly occur in canopy gaps. As image segmentations typically suffer from increased uncertainty at tile edges, we repeated the predictions with horizontal and vertical shifts of 256 pixels, which were subsequently aggregated using a majority vote.

The final model performance of $\mathrm{CNN}_{\mathrm{segment}}$ was assessed and compared to $\mathrm{CNN}_{\mathrm{window}}$ using the independent reference data (transects) obtained from the visual interpretation of the UAV orthoimages.

# 3 Results

For the $\text{CNN}_{\text{window}}$ method, F1-scores differed considerably across the tree species, while these differences were relatively consistent across the two orthoimages, i.e. $\text{Ortho}_{\text{July}}$ and $\text{Ortho}_{\text{September}}$(Fig. 4a, b). On a plot level, comparably high model performance (mean F1 > 0.6) was found for *Acer pseudoplatanus* and *Fraxinus exlcesior*, followed by the intermediate performance (mean F1-score 0.35-0.55) for *Aesculus hippocastanum*, *Sorbus aucuparia*, *Tilia platyphyllos*, *Betula pendula*, and *Carpinus betulus.* Low performance (mean F1-score < 0.35) was found for *Quercus petraea* , *Fagus sylvatica*, and *Prunus avium.* Averaged across species, there was a slight decrease in model performance from $\text{Ortho}_{\text{July}}$ with a mean F1-score of 0.44 to $\text{Ortho}_{\text{September}}$ with a mean F1-score of 0.4 (Fig. 4a, b). Note that $\text{Ortho}_{\text{July}}$ corresponded to the peak of the season, where leaves and canopies were still fully developed.

The $\text{CNN}_{\text{segment}}$ model performance across species was similar but generally higher compared to the $\text{CNN}_{\text{window}}$ method. For $\text{Ortho}_{\text{July}}$ F1-scores increased from 0.44 to 0.48 (Fig. 4a vs. c) and for $\text{Ortho}_{\text{September}}$, F1-scores increased from 0.40 to 0.46 (Fig. 4b vs. d).

We observed notable differences in model performance (mean F1) across different species mixtures, which are plots having one, two, or four species per plot (Fig. 5). For both $\text{CNN}_{\text{window}}$ and $\text{CNN}_{\text{segment}}$, the model performance strongly increased with lower number of species per plot (results for $\text{CNN}_{\text{window}}$ are given in the Appendix; Fig. A13).

The model performance of $\text{CNN}_{\text{segment}}$ exceeded the model performance of $\text{CNN}_{\text{window}}$ particularly in plots with increased number of species: For monocultures the relative increase in model performance (F1-score) amounted to 2.5%, in two species plots to 6.9%, and in plots with four species to 20.9% (averaged for $\text{Ortho}_{\text{July}}$ and $\text{Ortho}_{\text{September}}$). This increased performance can be attributed to the advantages of the encoder-decoder principle of the $\text{CNN}_{\text{segment}}$ method, enabling a pixel-wise and contextual prediction at the original resolution of the orthomosaics. These advantages are also visible in Fig. 6, where $\text{CNN}_{\text{segment}}$ resulted in more detailed and accurate tree species segmentations (particularly for plot 26 and 29).

The highest model performance for $\text{CNN}_{\text{segment}}$ was found in monoculture plots, where F1-scores > 0.5 was found for eight out ten species for both $\text{Ortho}_{\text{July}}$ and $\text{Ortho}_{\text{September}}$. A considerably lower performance for the July and September acquisition was found for *Prunus avium*, which may correspond to similarities in leaf and canopy structure with *Fagus sylvatica* and *Fraxinus excelsior* (a confusion matrix is given in the Appendix, Fig. A11). The decreased performance for *Carpinus betulus* and *Prunus avium* in $\text{Ortho}^{\text{September}}$ can be attributed to the very advanced senescence and leaf loss.

In addition to the increase in model performance, our analysis revealed that the prediction on orthoimagery using $\text{CNN}_{\text{segment}}$ only required 10% of the computation time compared to $\text{CNN}_{\text{window}}$. The duration of applying the models to the whole MyDiv orthomosaics covering an area of (3.02 hectare; 0.22 cm ground sampling distance) took approximately 27.05 hours with $\text{CNN}_{\text{segment}}$ and 264.88 hours with $\text{CNN}_{\text{window}}$ (NVIDIA A6000 with 48 GB RAM).

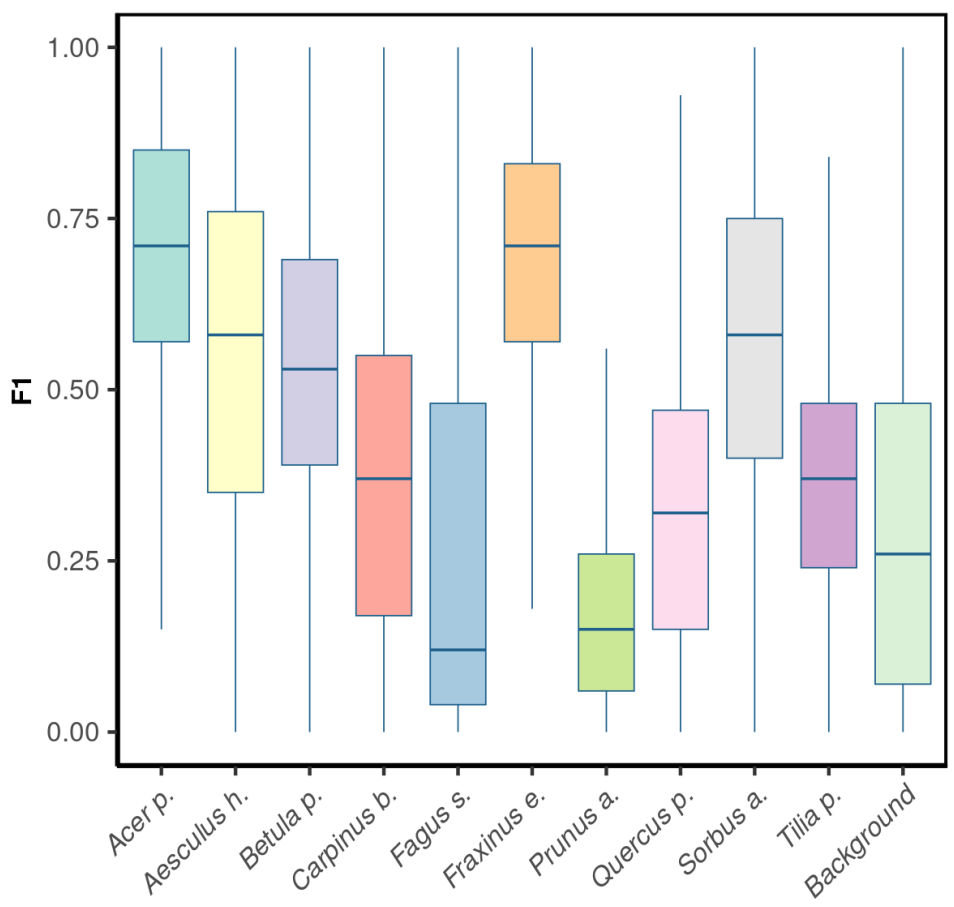

(a) F1-scores for $CNN_{window}$ on $Ortho_{July}$ (mean 0.44).

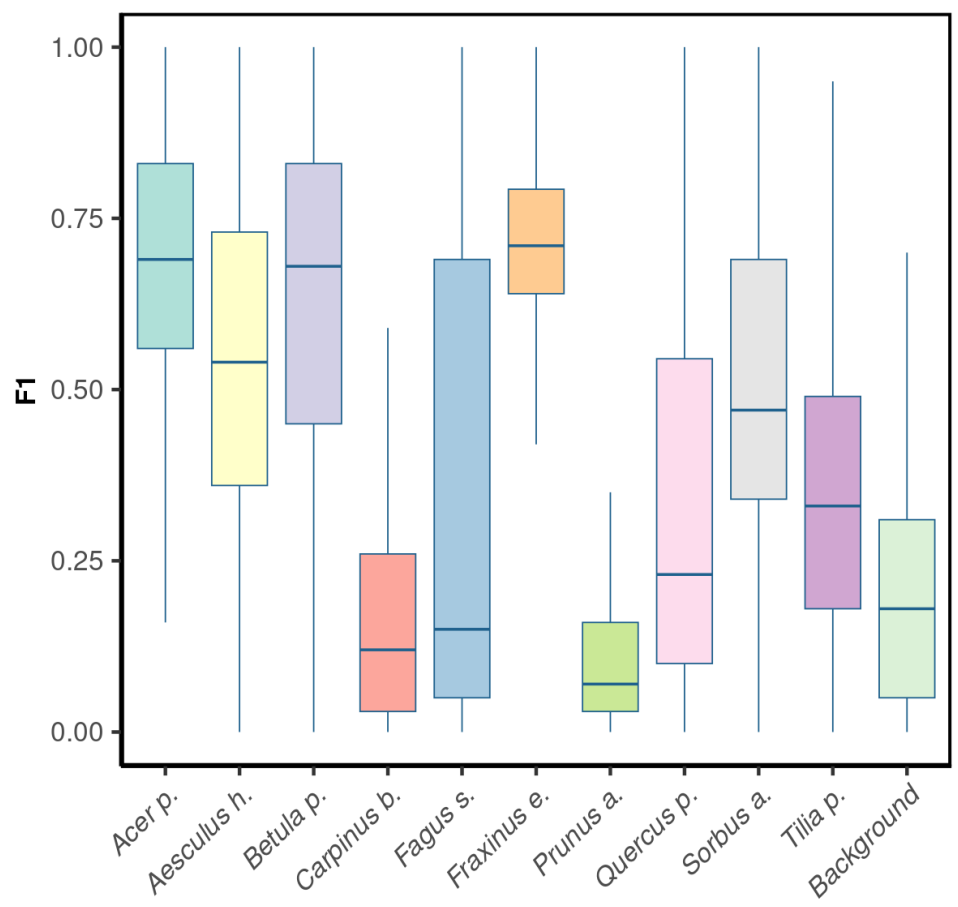

(b) F1-scores of$CNN_{window}$ on $Ortho_{September}$ (mean 0.42).

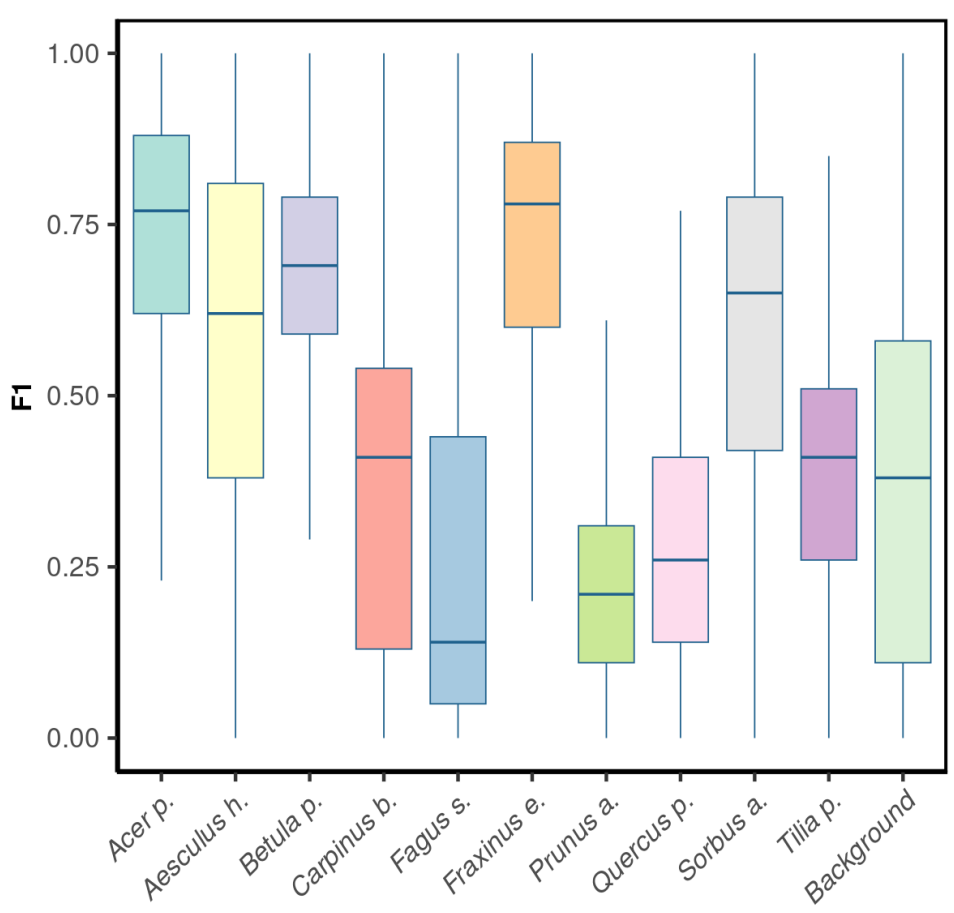

(c) F1-scores of $CNN_{segment}$ on $Ortho_{July}$ (mean 0.48).

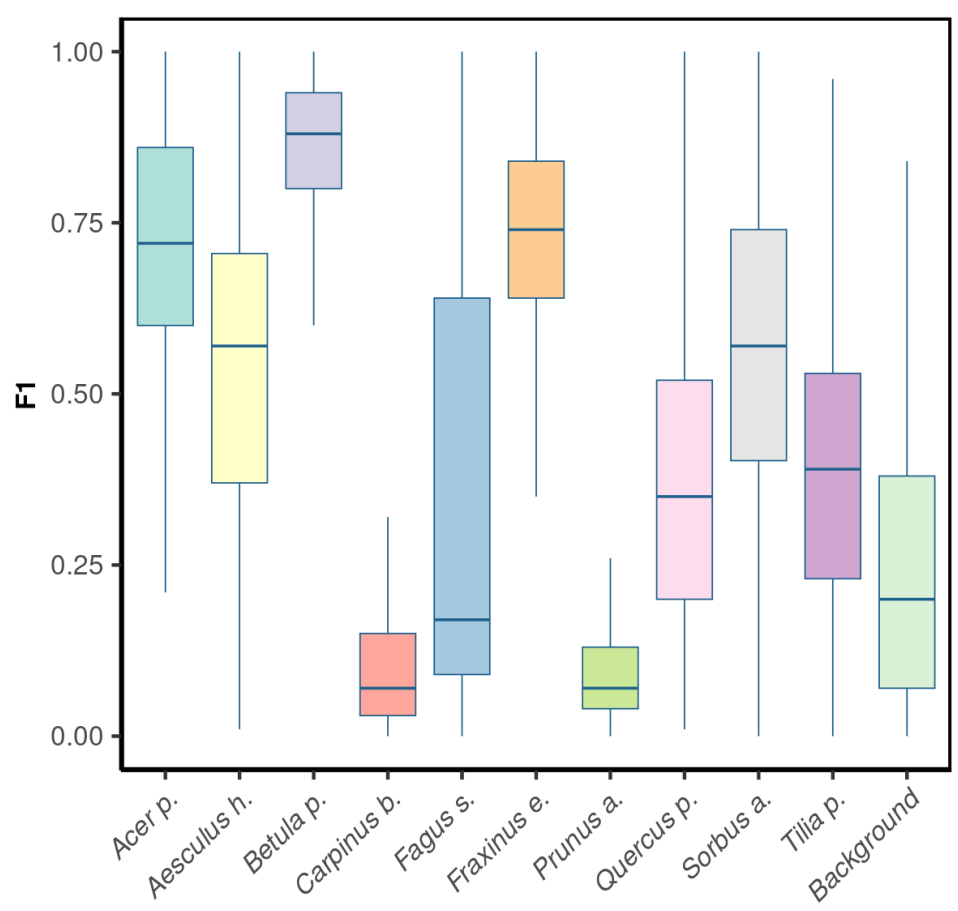

(d) F1-scores of $CNN_{segment}$ on $Ortho_{September}$ (mean 0.46).

Figure 4: F1-scores by tree species and background class for $Ortho_{July}$ and $Ortho_{September}$ derived from $CNN_{window}$ and $CNN_{segment}$.

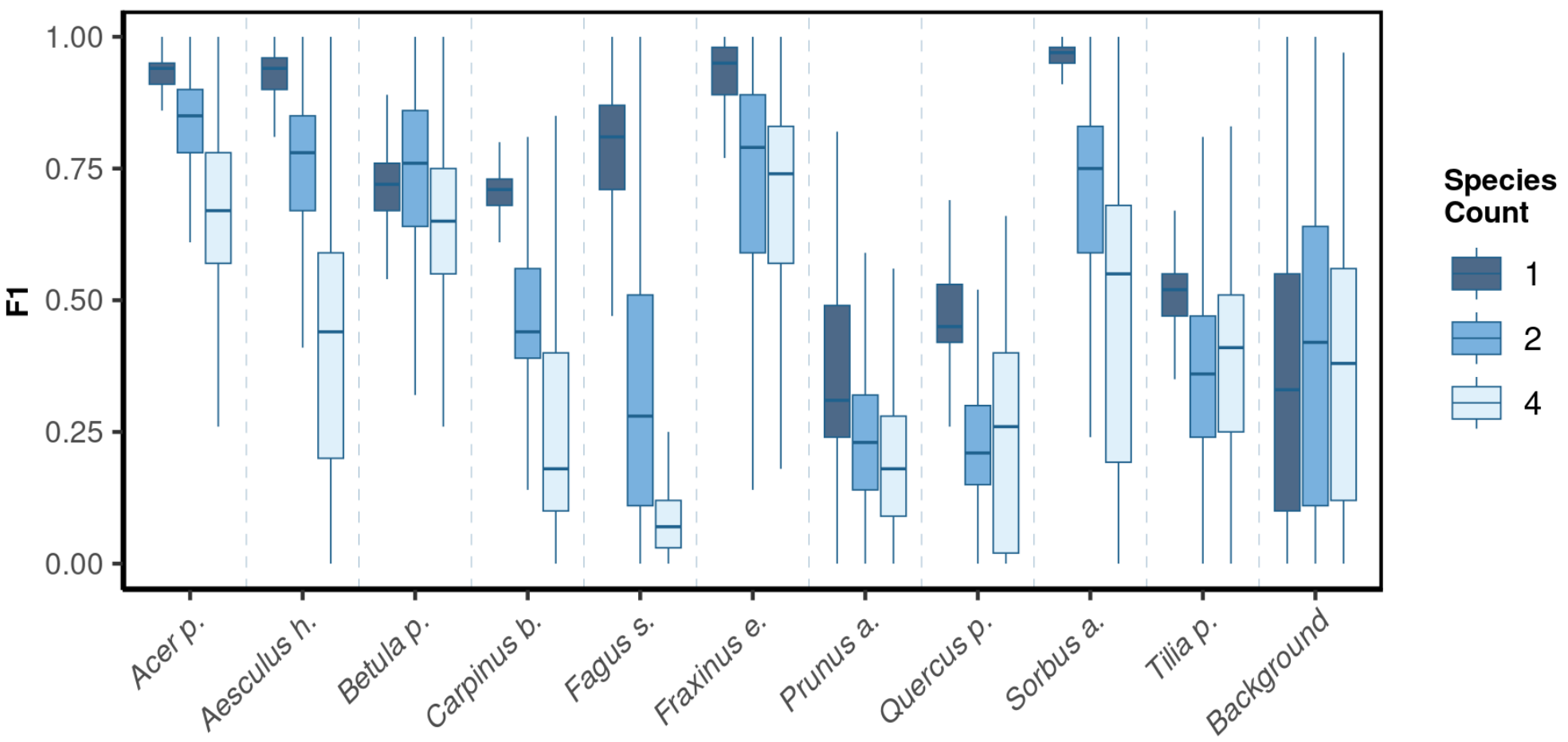

(a) Performance across species mixtures (F1-scores) on $\text{Ortho}_{\text{July}}$. Mean F1-scores: 1 species (0.51), 2 species (0.44), 4 species (0.41)

.

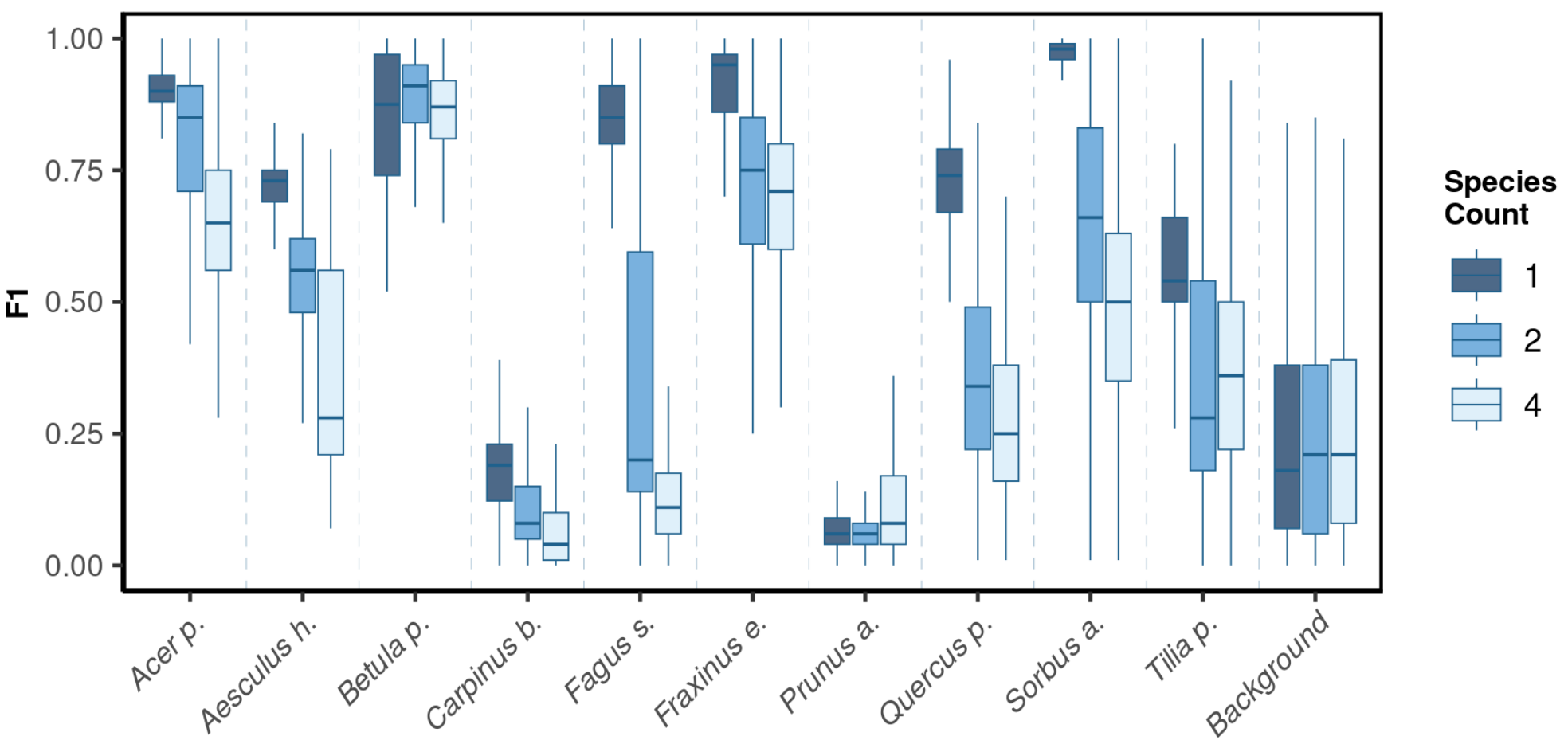

(b) Performance across species mixtures (F1-scores) on $\text{Ortho}_{\text{September}}$. Mean F1-scores: 1 species (0.58), 2 species (0.51), 4 species (0.42)

Figure 5: The model performance (F1-score) of the $\text{CNN}_{\text{segment}}$ model across a gradient of canopy complexity in $\text{Ortho}_{\text{July}}$ and $\text{Ortho}_{\text{September}}$. F1-scores decrease with increasing canopy complexity in plots

.

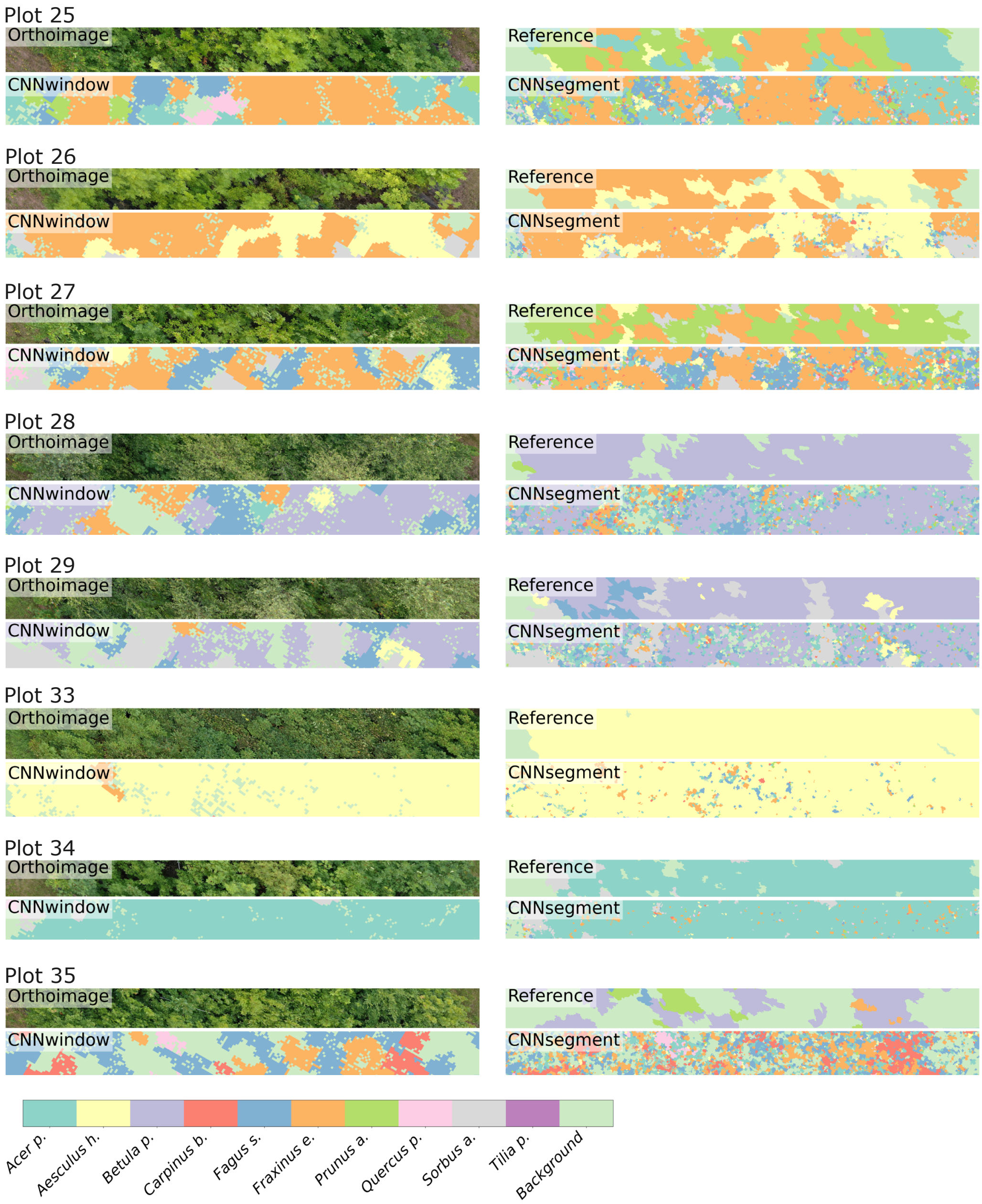

Figure 6: Transects of 2 m by 20 m of selected plots, including the orthoimage, the reference, $CNN_{window}$ predictions, and $CNN_{segment}$ predictions. Visualizations for the remaining plots are given in the Appendix (Section A1.1).

# 4 Discussion

## 4.1 Filtering of citizen science data for drone-related applications

To achieve better correspondence between plant features visible in the citizen science photographs and the UAV images, we filtered the crowd-sourced photographs based on their acquisition distance (less than 0.3 m or greater than 15 m) to exclude macro and landscape photographs. Moreover, we excluded photographs that predominantly display tree trunks, facilitating a foliage-centric perspective as intrinsic to high-resolution UAV images (Fig. 3). In the future, more criteria may be used for filtering citizen science imagery, including metadata (labels) on the presence of specific plant organs within an image (e.g., fruits, flowers) as provided as a by-product by some citizen science plant identification apps (e.g., Pl@ntNet).

## 4.2 The creation of segmentation masks from simple image labels

One of the challenges of generating segmentation masks for the encoder-decoder method ($CNN_{segment}$) with the proposed workflow may be error propagation between the different steps. Firstly, the CNN image classification trained on the citizen science data has variyin uncertainty for the different species, resulting from noisy citizen science observations or limitations to identify some species solely by photographs (Van Horn et al., 2018). Secondly, the moving window approach ($CNN_{window}$), which predicts one species for an entire tile, may be too coarse to resemble very complex canopies (e.g., in highly diverse plant communities). However, although the fact that the segmentation labels created with the $CNN_{window}$ approach are partially relatively inaccurate (Fig 4a, 6), we found that the $CNN_{segment}$ procedure indeed resulted in higher performance than the $CNN_{window}$ procedure. This is in line with other studies (Kattenborn et al., 2021; Cloutier et al., 2023; Schiller et al., 2021) reporting that deep learning-based pattern recognition can partially overcome noise labels, whereas the intentional use of noisy reference data, also known as weakly-supervised learning, is generally very promising in the absence of high-quality labels (Zhou, 2018). Here, we filtered the training data (masks) for regions where we expect extreme noise levels, that is, for tiles where none of the classes exceeded a relative cover of 30%. These regions were, according to our observation, often canopy gaps and shadowed areas, where one naturally expects lower model performance due to less distinct species-specific textures (Lopatin et al., 2019; Milas et al., 2017; De Sa et al., 2018).

The enhanced segmentation performance of the $CNN_{segment}$ approach compared to $CNN_{window}$ can be attributed to the spatially explicit and finer-resolved predictions of the U-Net segmentation algorithm (encoder-decoder principle), enabling to segment the tree species at the native resolution of the orthoimagery. Particularly, for plots with more species (two or four) the encoder-decoder segmentation approach resulted in improved prediction results compared to the $CNN_{window}$ method in plots with more species (two or four) and hence, more complex canopies. Thus, the presented two-step approach of creating segmentation masks from simple class labels $CNN_{window}$, as provided by iNaturalist and Pl@ntNet platforms, can indeed be used to create segmentation masks required for state-of-the-art image analysis methods

($CNN_{segment}$) and thereby result in higher value for remote sensing applications. The increased value of these segmentation masks enables the training of algorithms with higher performance in species recognition. It greatly enhances the efficiency of applying the models on orthoimagery (factor of approximately ten). Especially for recurrent applications, such as monitoring or large-scale undertakings, the two-step approach involving the creation of segmentation masks and encoder-decoder architectures is recommended.

### 4.3 The role of canopy complexity

Overall, the segmentation performance declined with increasing species richness per plot. We expect that this can mainly be attributed to the small size of individual trees at the MyDiv site, where in high species mixtures, there is a lower chance that a 512 × 512 pixel tile includes clearly visible species-specific leaf and branching patterns. This also explains why, in particular, trees with lower relative canopy height (e.g., *Quercus petrea* and *Fagus sylvatica* were less likely to be accurately segmented in species mixtures. The observed effect of canopy complexity is in line with previous findings from Soltani et al. (2022); Lopatin et al. (2017); Fassnacht et al. (2016); Fricker et al. (2019), where smaller patches of individual species were less likely to be accurately detected. Visual inspection also confirmed that false predictions were more likely at canopy edges between different tree species (Fig. 6). However, it should be noted that the small-scaled canopy complexity of the plots used here is exceptionally high (Fig. 3). Most tree crowns in the MyDiv experiment do not exceed a diameter of 1.5 m, and the transition among tree crowns of multiple species is often very fuzzy. Thus, we expect reduced performance in canopy transitions to be less relevant in real-world settings, where tree species appear in more extensive, homogeneous patches and where individual crowns are commonly larger. Thus, the model performance in these species mixtures can be interpreted as a rather conservative estimate. The results obtained for the monocultures might be more representative in terms of real-world applications, as mature trees in temperate forests typically have crown diameters 5 to 20 times larger. Application tests of the presented approach in real forests are desirable. However, acquiring such a dataset is a logistical challenge since temperate forest stands commonly do not feature a comparably high and balanced occurrence of that many tree species.

### 4.4 Spatial resolution of the UAV imagery is key

According to the results obtained in the monocultures, The $CNN_{segment}$ model successfully classified seven out of ten tree species (F1 > 0.7). The lower F1-scores for *Quercus petrea* (mean F1 0.57), *Prunus avium*(mean F1 0.2), *Tilia platyphyllos*(mean F1 0.53) may result from the spectral and morphological similarity at the current spatial resolution of the UAV imagery (0.22 cm)(Fig. 3). Hence, there was a tendency that these species were often confused with each other (see confusion matrices in Appendix A1.2). Such confusion among plants with a similar appearance was confirmed by other studies (Cloutier et al., 2023; Schiefer et al., 2020, e.g.) and matches our experience from the generation of reference data via visual

interpretation, where a separation between these species was sometimes challenging. Initial CNN-based segmentation attempts (results not shown) in the preparation of this study were based on an orthoimage of 0.3 cm instead of 0.22 cm resolution, resulting in clearly lower model performances. This aligns with the reported importance of spatial resolution of UAV imagery for CNN segmentation of earlier studies (Schiefer et al., 2020; Schmitt et al., 2020; Ma et al., 2019; G. Braga et al., 2020). Thus, while the current orthoimages with 0.22 cm resolution delivered promising results, further increasing the spatial resolution might be very promising for species where characteristic leaf forms can only be visualized at fine spatial resolutions.

### 4.5 Model transferability across seasons and orthoimage acquisition properties

The diversity of human behavior and electronic devices makes citizen science-based plant photographs very heterogeneous. This can be a challenge for deep learning applications, such as species recognition or plant trait characterization (Schiller et al., 2021; Van Horn et al., 2021; van Der Velde et al., 2023; Affouard et al., 2017), where models have to identify features that hold across various viewing angles, distances, or illumination conditions. However, this heterogeneity might also be of great value, given that citizens depict the appearance of plants under various site, environmental, and phenological conditions. This, in turn, offers a unique setting for training models that are generic and transferable across these conditions. Here, we evaluated the transferability of our models across different data sets by applying them to two orthoimages acquired in different seasons (peak of growing season and autumn). Both the $\text{CNN}_{\text{window}}$ and $\text{CNN}_{\text{segment}}$ models could identify deciduous tree species in the orthoimages with surprising accuracies, suggesting that the models are transferable to different conditions.

### 4.6 Outlook

Overall, our results indeed highlight the value of citizen science photographs with simple class labels to create training data for state-of-the-art segmentation approaches. A great advantage of this citizen science-based approach is that it does not require commonly costly training data obtained from visual interpretation or field surveys (here, we only acquired reference data for validating the procedure). This particularly highlights the potential of citizen science data for applications where many species are of interest, such as biodiversity-related monitoring applications (Chandler et al., 2017; Johnston et al., 2023). In this regard, data or models of species-recognition platforms that incorporate excessive amounts of plant species and respective imagery are very promising, including iNaturalist (Boone and Basille, 2019), Pl@ntNet (Affouard et al., 2017), ObsIdentify (Molls, 2021) or FloraIncognita (Mäder et al., 2021). Yet, based on the current and the precursor study (Soltani et al., 2022), we expect that a pre-selection of citizen science photograph databases considering images more representative of the common UAV-based perspective is required to unleash the potential of this heterogeneous data.

## 5 Conclusion

The transfer learning approach presented here demonstrates the value of freely available crowd-sourced plant photographs for remote sensing studies. This heterogeneous dataset can provide valuable training data for transferable CNN-based segmentation models. Here, this potential was highlighted in a very complex task, i.e., the differentiation of multiple temperate deciduous tree species in mixed vegetation stands with a complex structutre. The presented two-step approach demonstrated how we can transfer and harness generic knowledge gathered by citizens on how plants 'look' to the bird perspective of high-resolution drone imagery. The presented moving window approach overcomes the limitation of citizen science-based photographs having only simple species labels. The segmentation maps derived from an image classification model applied in a moving window setting can be harnessed to create segmentation masks for encoder-decoder-type segmentation models. The latter does not only enable higher accuracies in species segmentation but is also considerably more efficient. By building on the effort of thousands of citizens, this framework enables the mapping of plant species without any training data obtained from visual interpretation or ground-based field surveys. Due to the excessive amounts of plant photographs acquired in different conditions, such models can be assumed to have a large transferability.

## 6 Data and code availability

The code used in this study is publicly accessible via our GitHub repository at https://github.com/salimsoltani28/CrowdVision2TreeSegment. The data supporting the findings of this research is available on Zonodo at https://zenodo.org/uploads/10019552.

## 7 Declaration of competing interest

The authors declare that they have no known competing financial interests or personal relationships that could have appeared to influence the work reported in this paper.

## 8 Acknowledgements

SS and TK acknowledge funding by the German Research Foundation (DFG) under the project BigPlantSens (Assessing the Synergies of Big Data and Deep Learning for the Remote Sensing of Plant Species; Project number 444524904) and PANOPS (Revealing Earth´s plant functional diversity with citizen science; project number 504978936). SS and HF acknowledge financial support by the Federal Ministry of Education and Research of Germany (BMBF) and by the Saechsische Staatsministerium für Wissenschaft, Kultur und Tourismus in the program Center of Excellence for AI-research "Center for Scalable Data Analytics and Artificial Intelligence Dresden/Leipzig", project identification number: ScaDS.AI. NE and OF acknowledge funding by the Deutsche Forschungsgemeinschaft DFG (German Centre for In-

tegrative Biodiversity Research, FZT118; and Gottfried Wilhelm Leibniz Prize, Ei 862/29-1). Moreover, we acknowledge support from Leipzig University for Open Access Publishing.

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

# A Appendix

## A1.1 Prediction maps

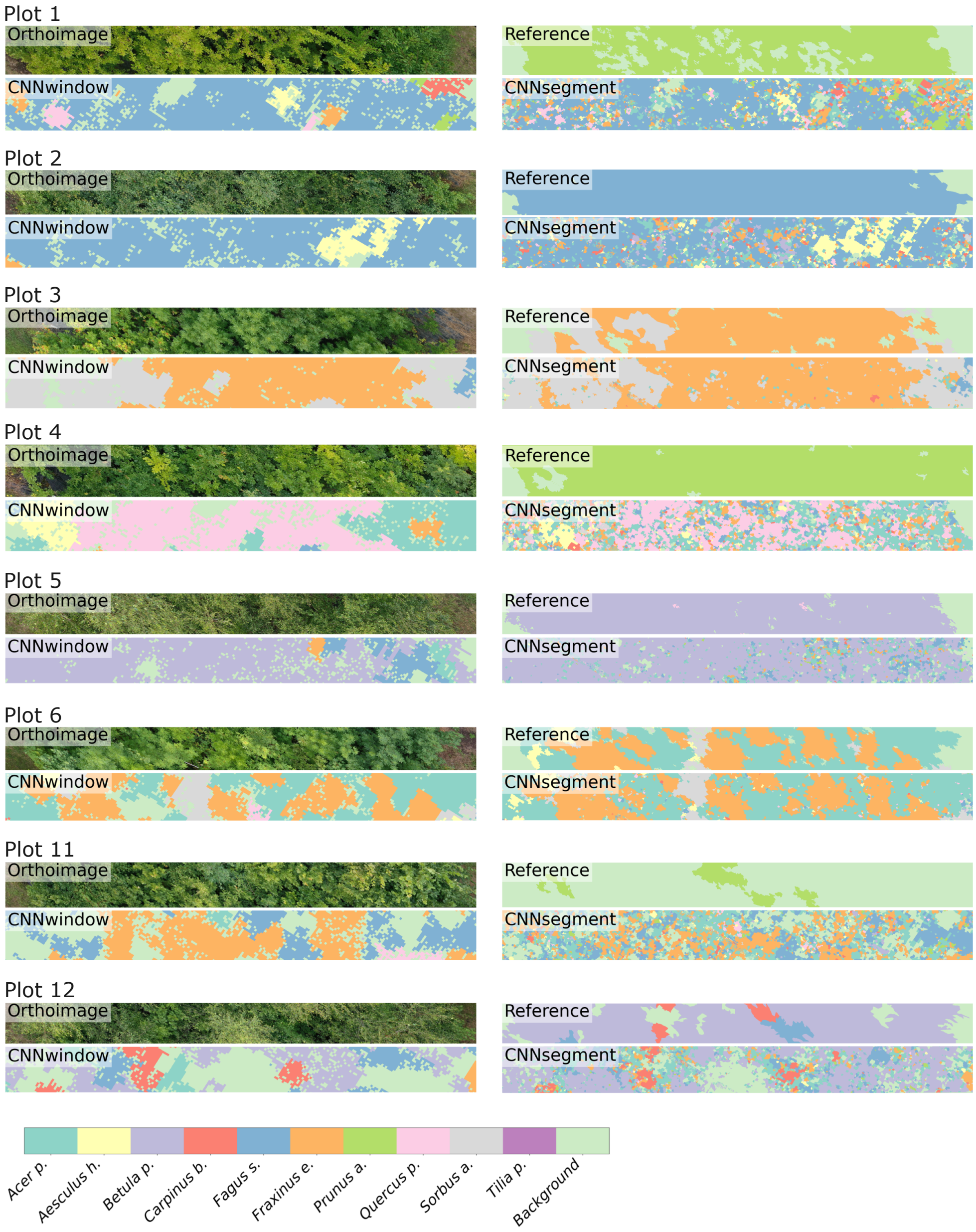

Figure A1: Transects of 2 m by 20 m of selected plots, including the orthoimage, the reference, $\mathrm{CNN_{window}}$ predictions, and $\mathrm{CNN_{segment}}$ predictions.

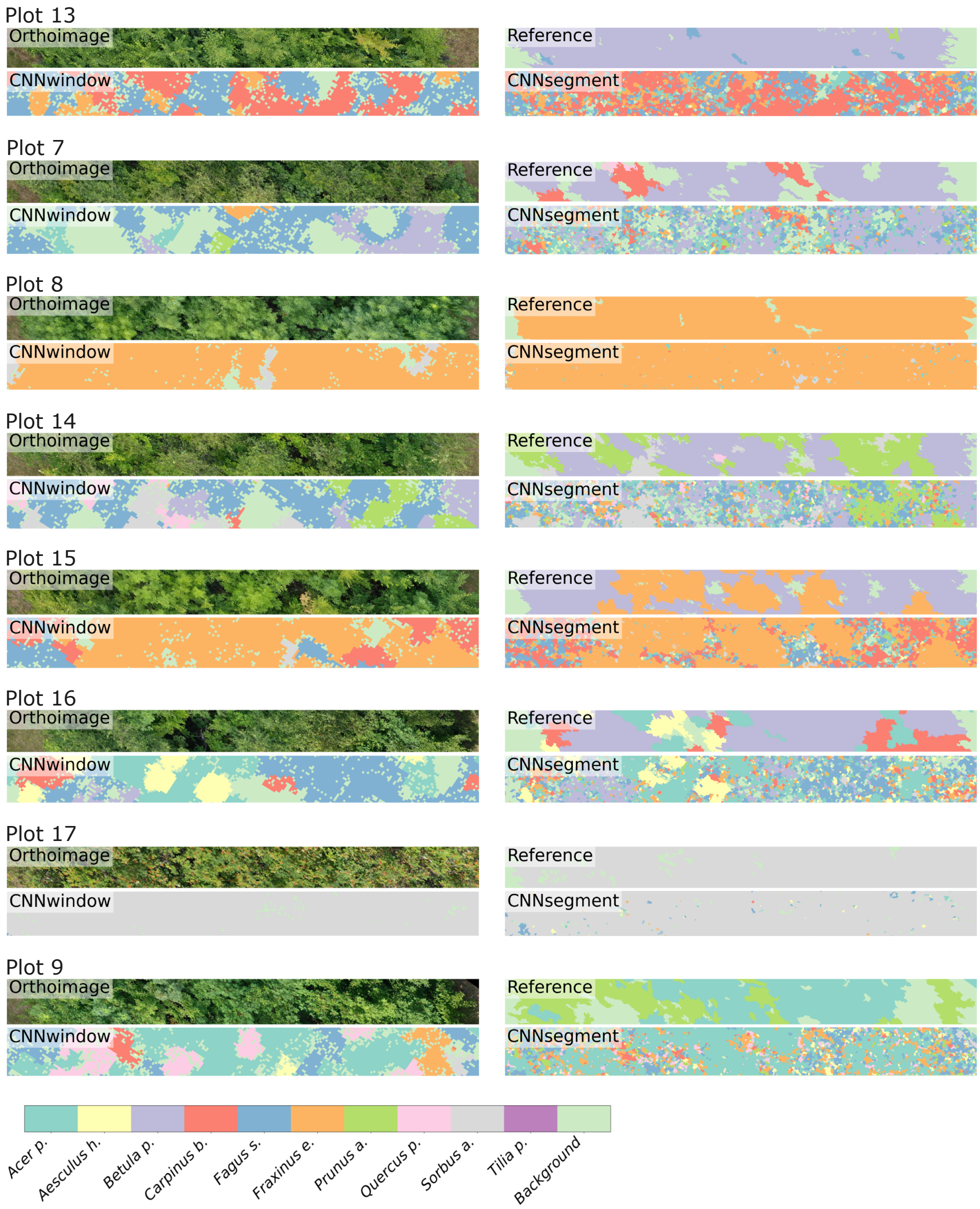

Figure A2: Transects of 2 m by 20 m of selected plots, including the orthoimage, the reference, $\text{CNN}_{\text{window}}$ predictions, and $\text{CNN}_{\text{segment}}$ predictions.

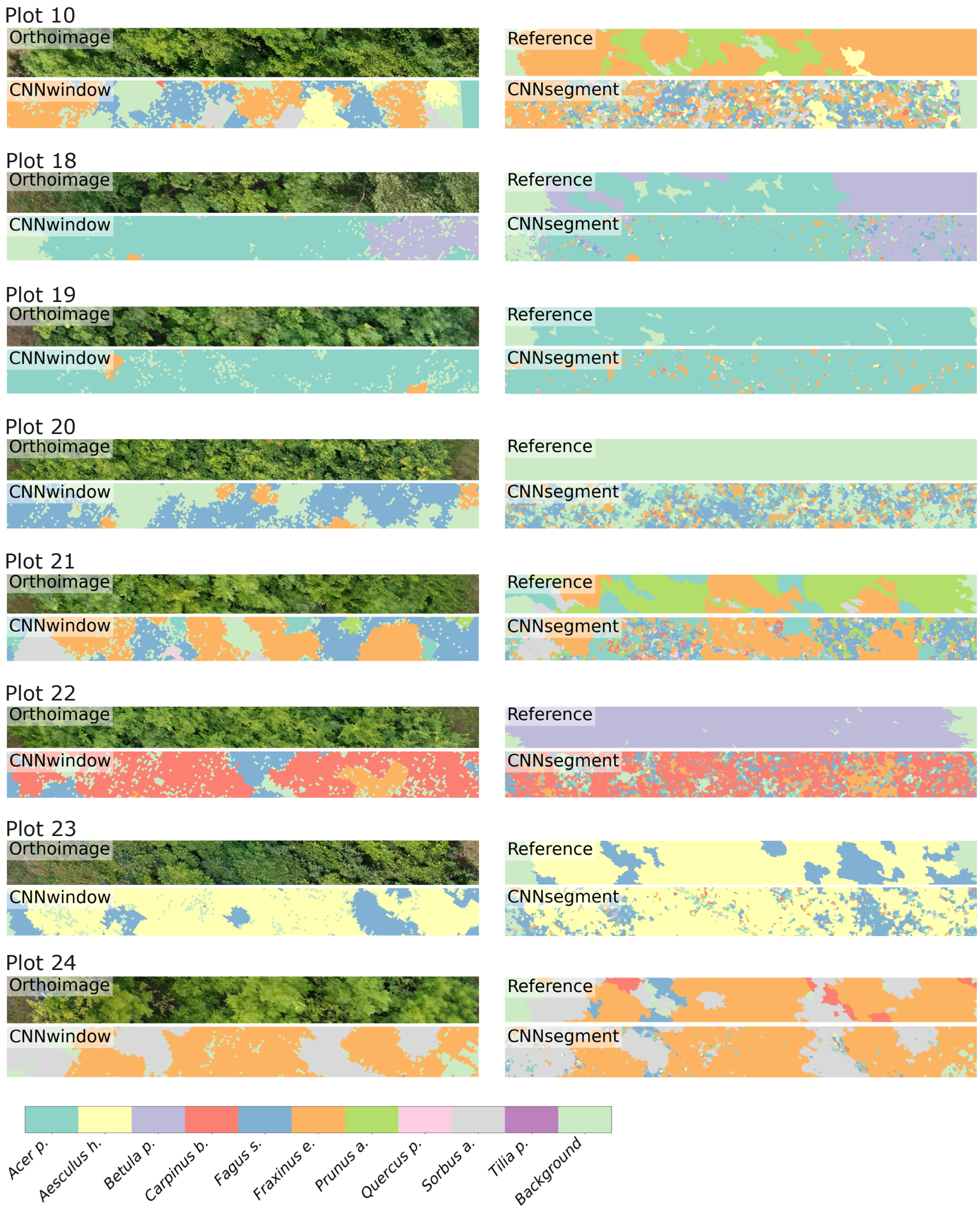

Figure A3: Transects of 2 m by 20 m of selected plots, including the orthoimage, the reference, $CNN_{window}$ predictions, and $CNN_{segment}$ predictions.

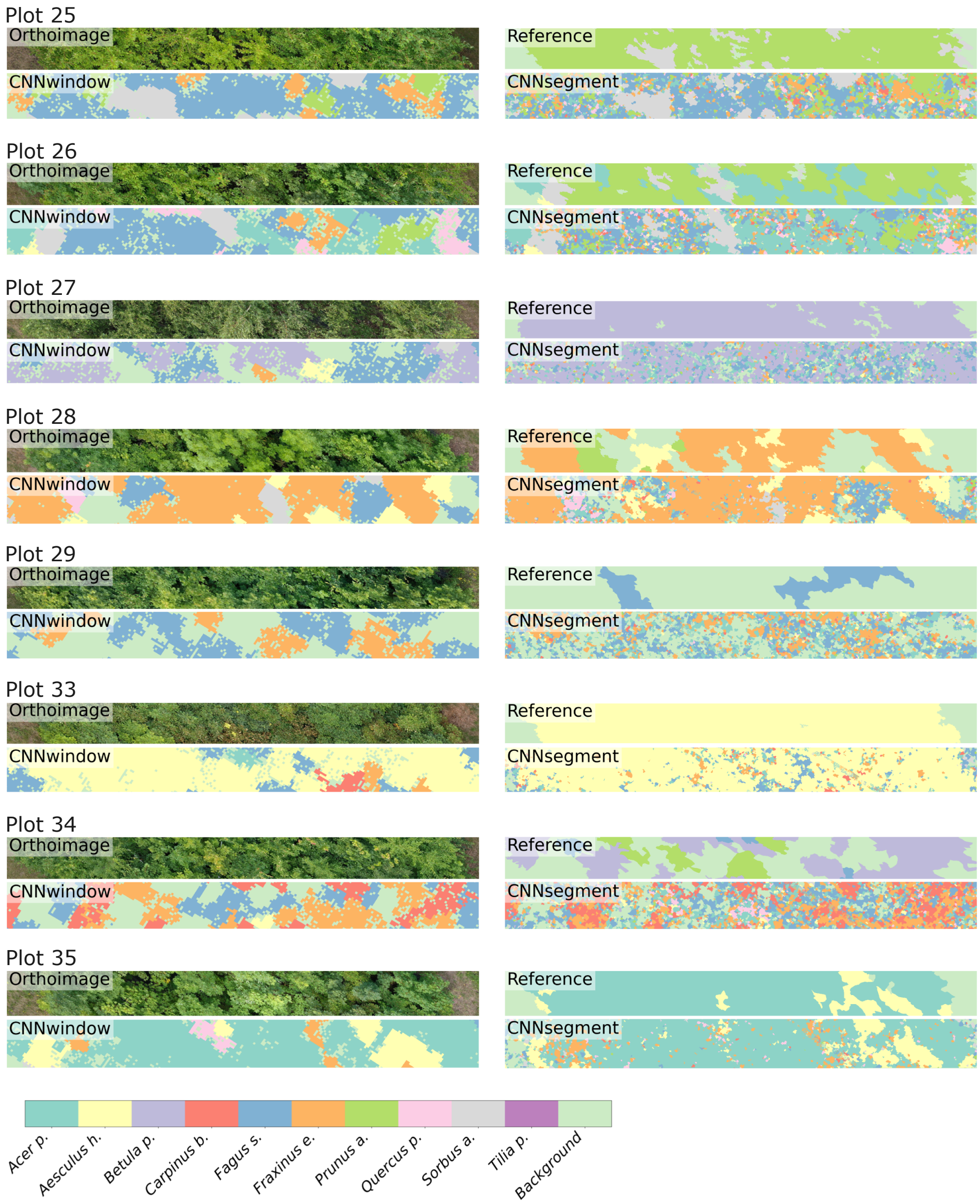

Figure A4: Transects of 2 m by 20 m of selected plots, including the orthoimage, the reference, $\mathrm{CNN}_{\mathrm{window}}$ predictions, and $\mathrm{CNN}_{\mathrm{segment}}$ predictions.

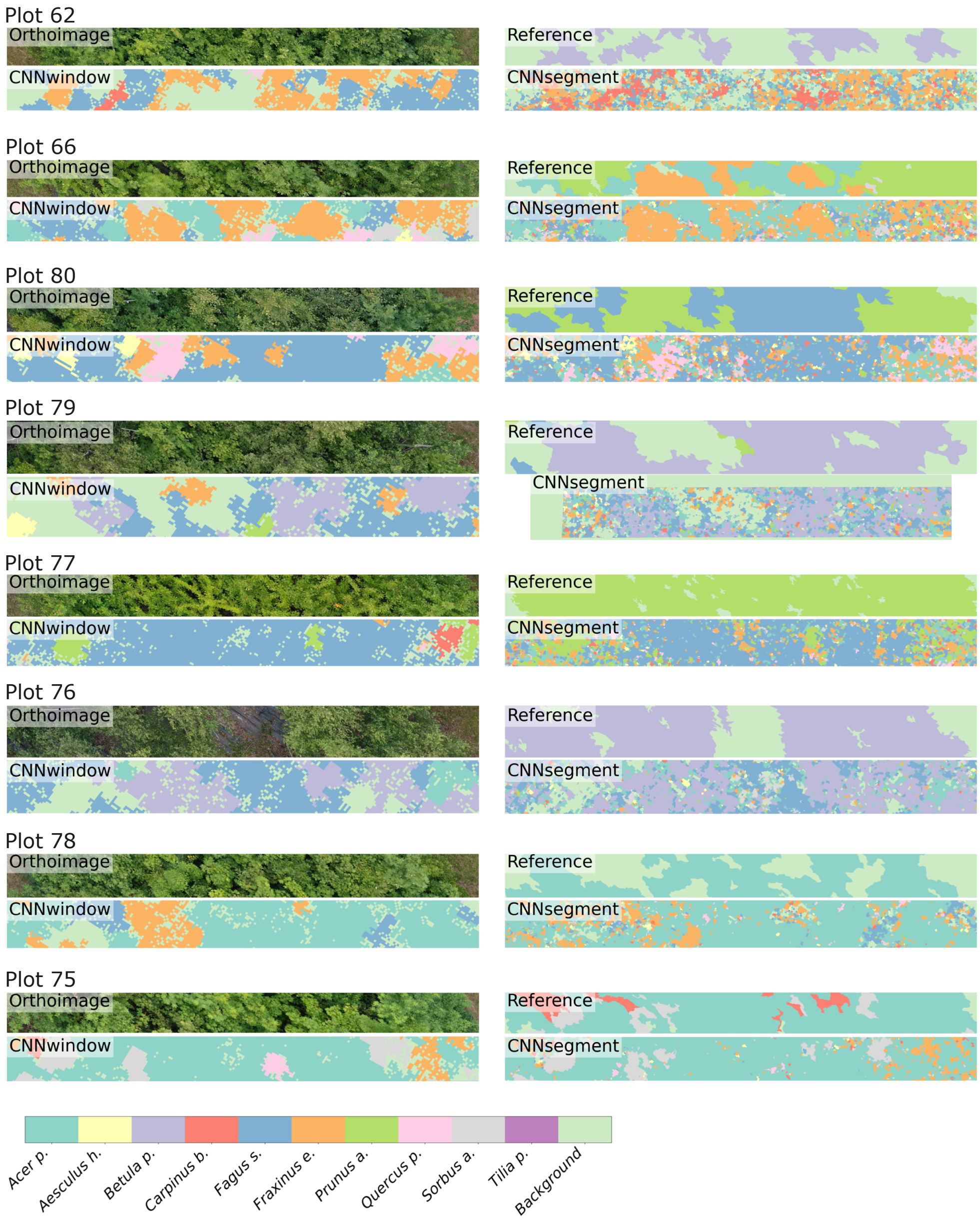

Figure A5: Transects of 2 m by 20 m of selected plots, including the orthoimage, the reference, $\mathrm{CNN}_{\mathrm{window}}$ predictions, and $\mathrm{CNN}_{\mathrm{segment}}$ predictions.

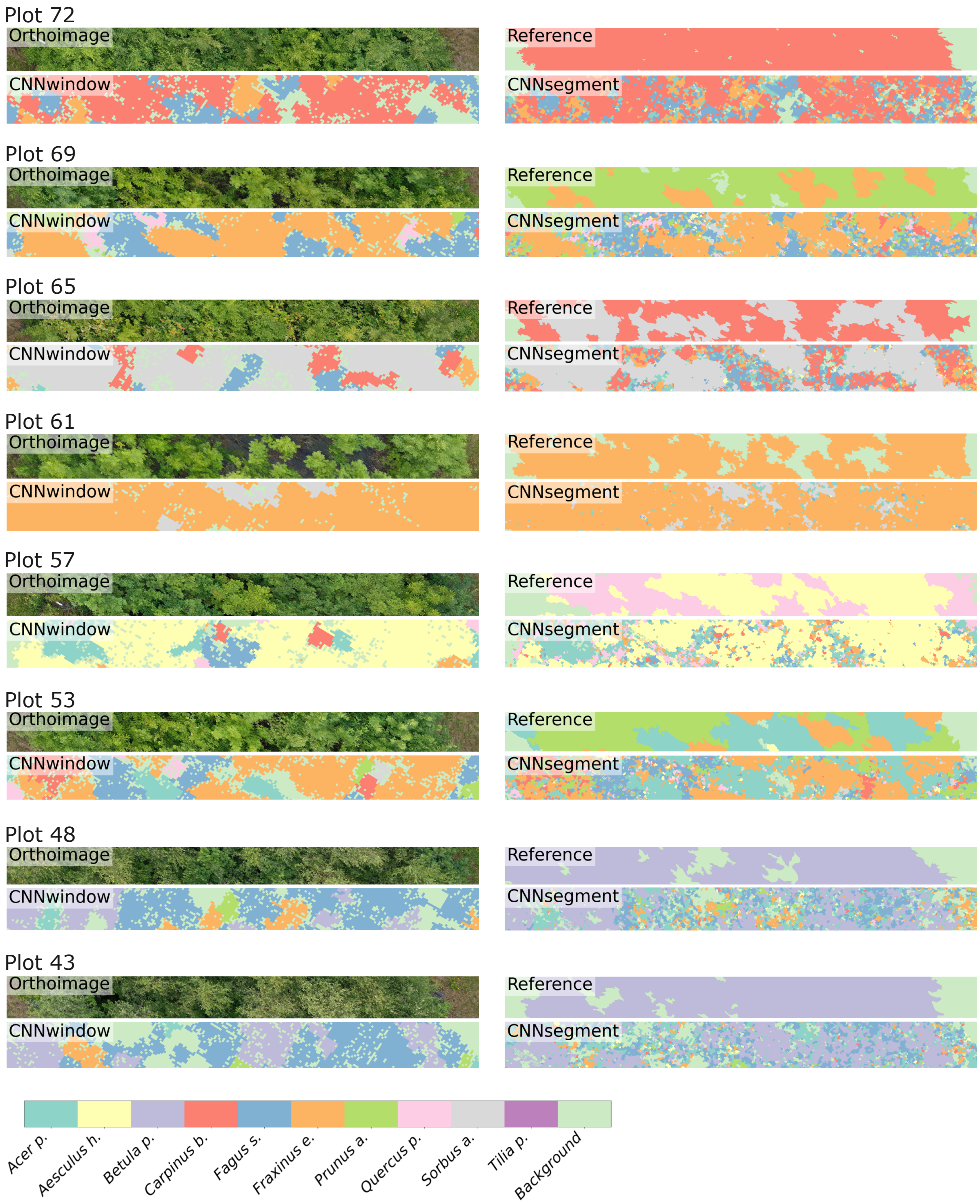

Figure A6: Transects of 2 m by 20 m of selected plots, including the orthoimage, the reference, $CNN_{window}$ predictions, and $CNN_{segment}$ predictions.

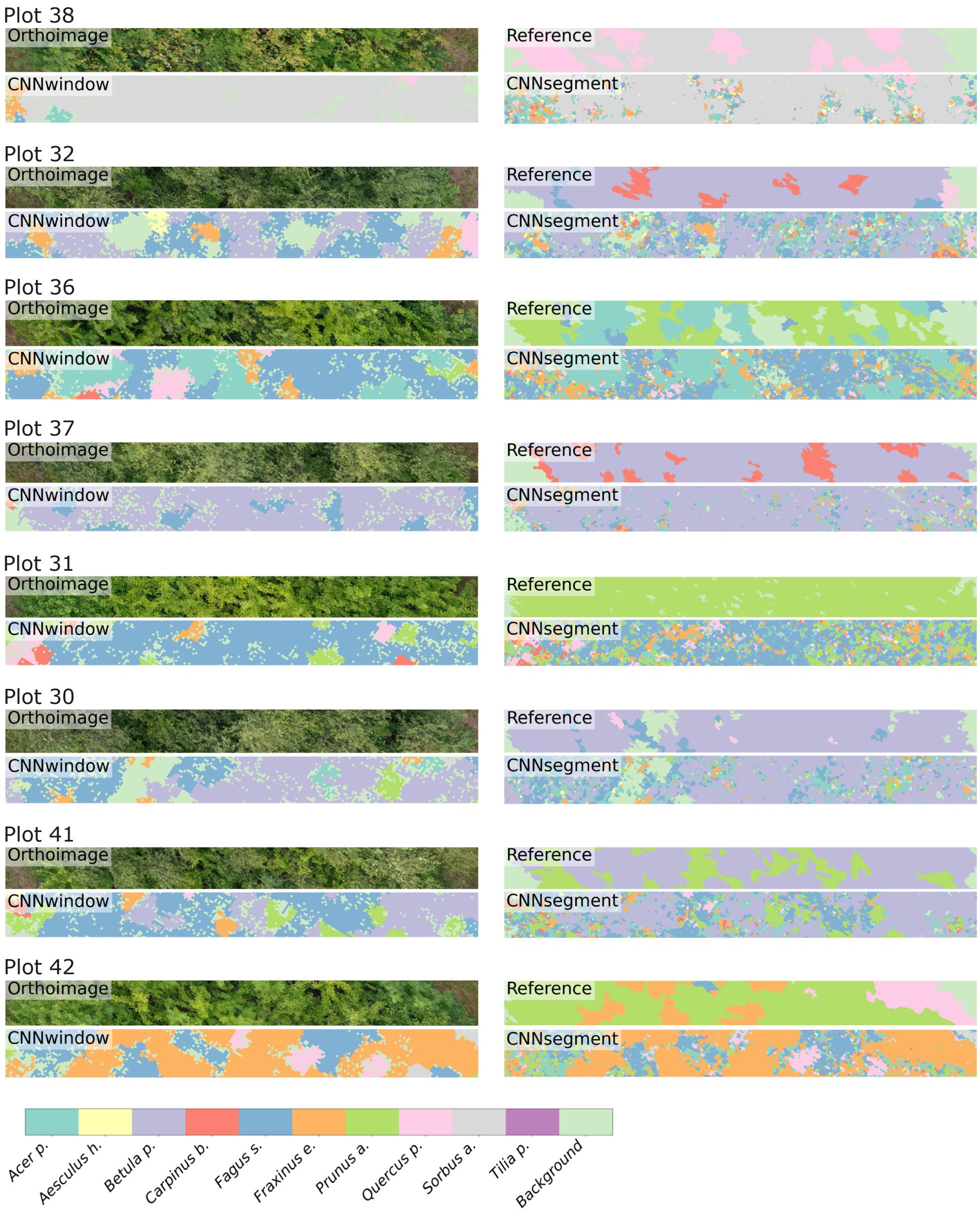

Figure A7: Transects of 2 m by 20 m of selected plots, including the orthoimage, the reference, $\text{CNN}_{\text{window}}$ predictions, and $\text{CNN}_{\text{segment}}$ predictions.

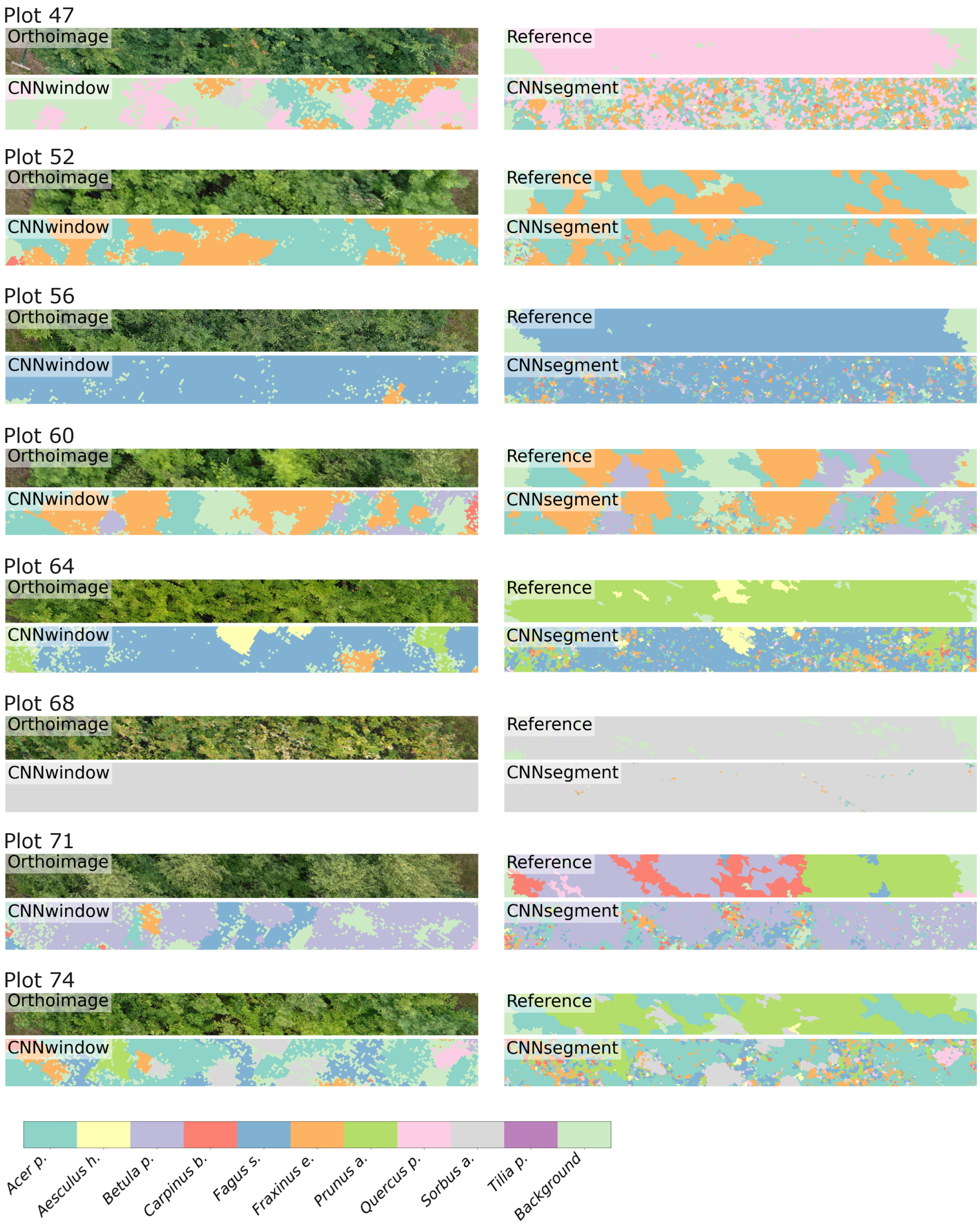

Figure A8: Transects of 2 m by 20 m of selected plots, including the orthoimage, the reference, $CNN_{window}$ predictions, and $CNN_{segment}$ predictions.

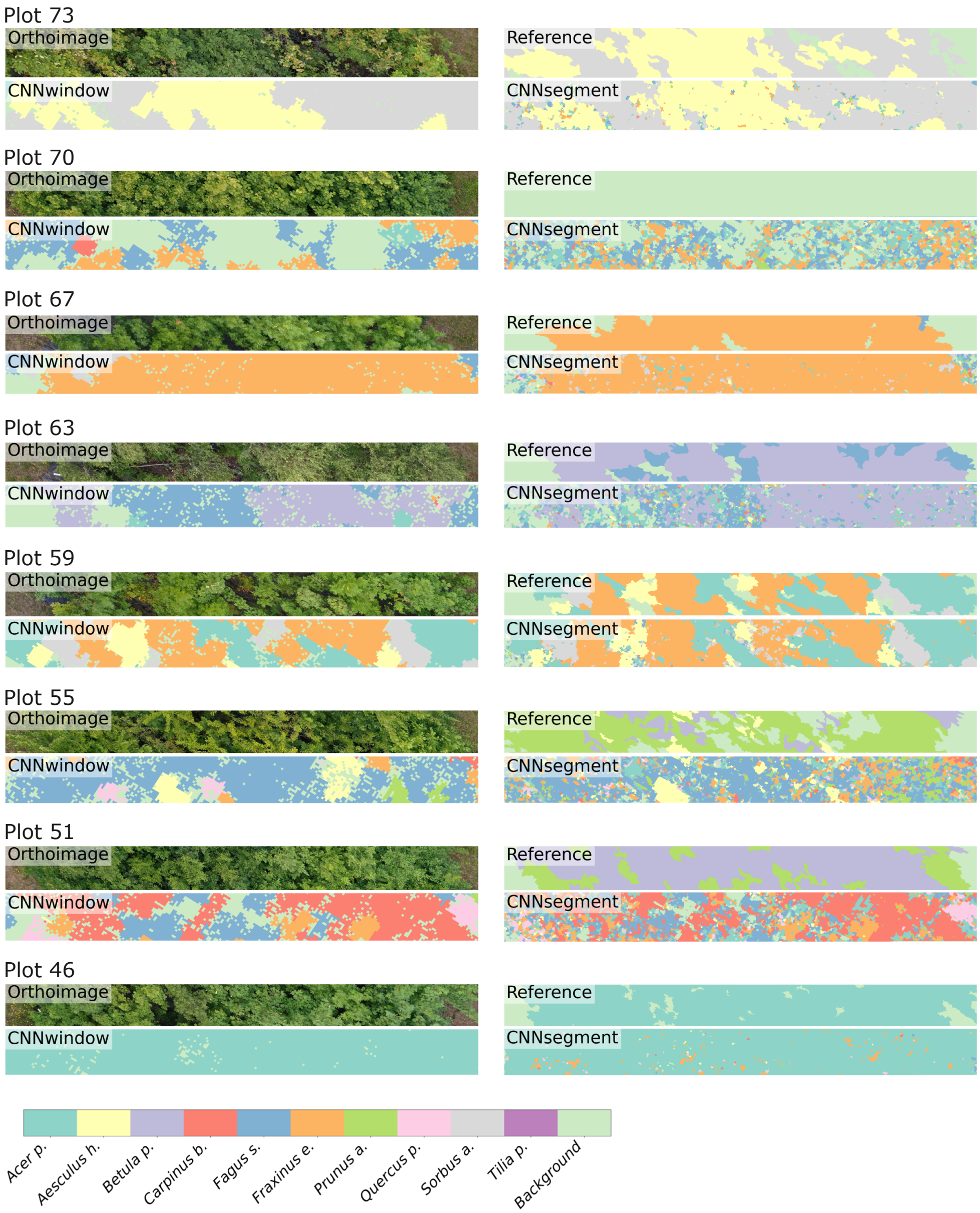

Figure A9: Transects of 2 m by 20 m of selected plots, including the orthoimage, the reference, $CNN_{window}$ predictions, and $CNN_{segment}$ predictions.

### A1.2 Confusion Matrix

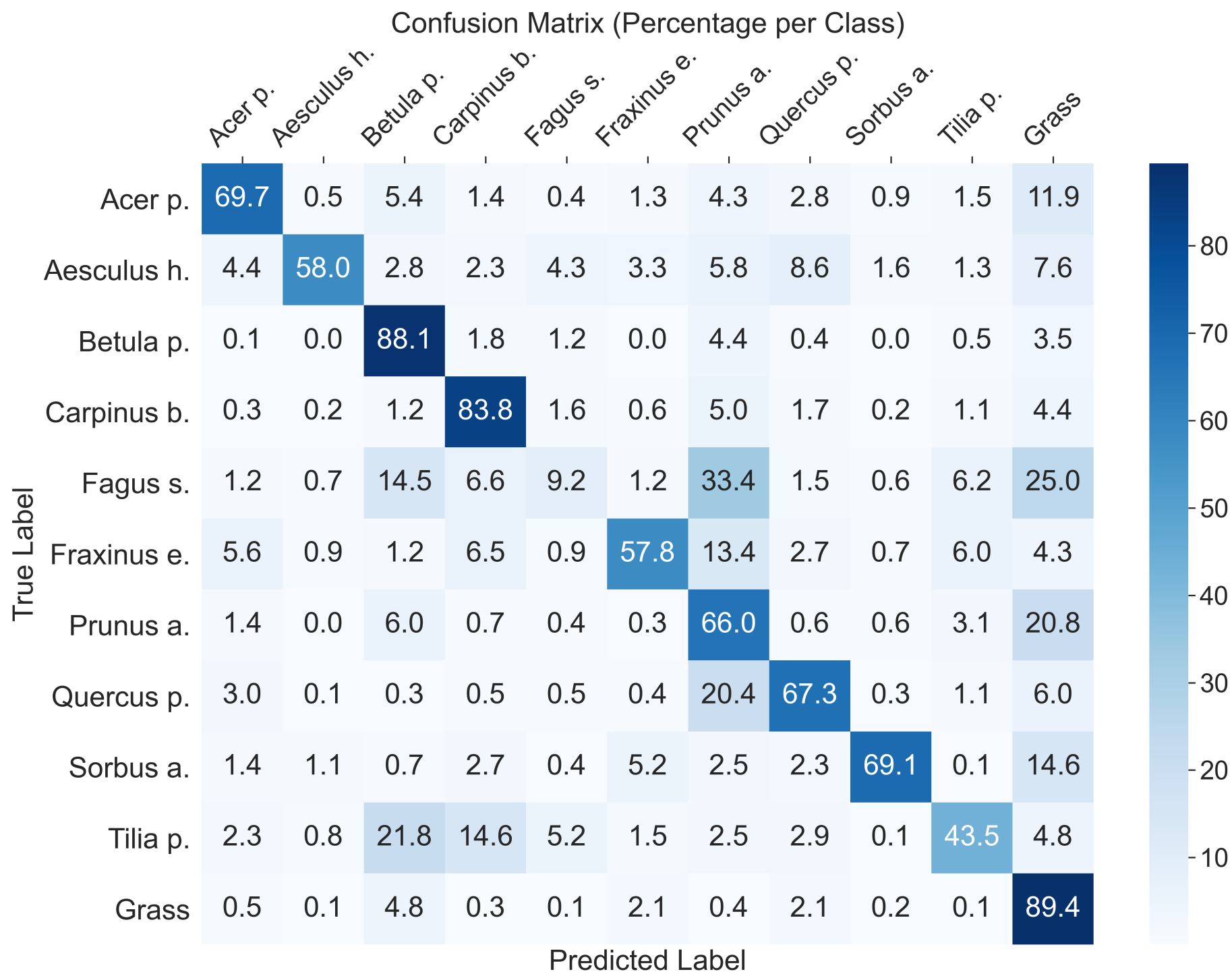

Figure A10: Normalized Confusion Matrix of the CNNsegment model applied to $\text{Ortho}_{\text{September}}$

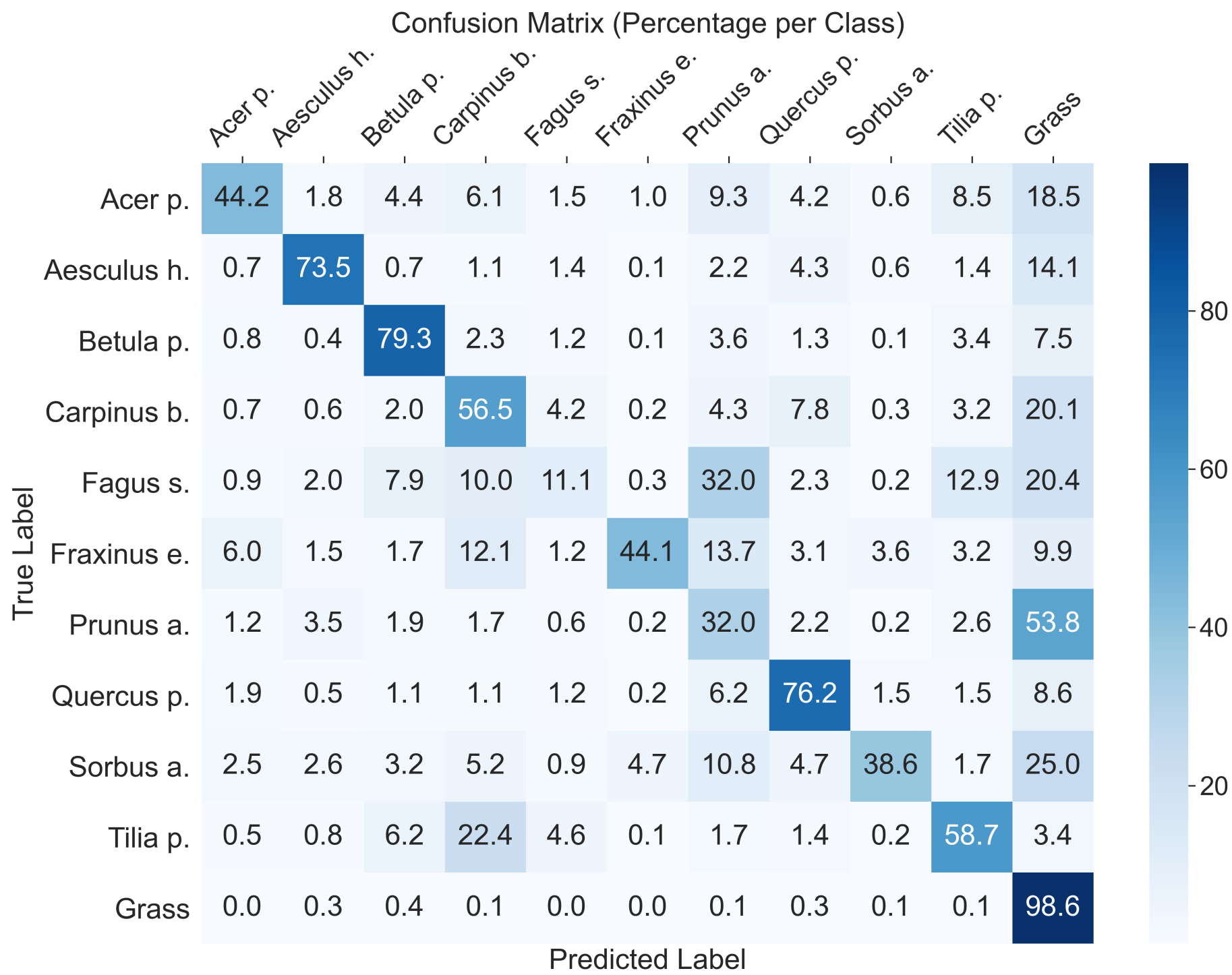

Figure A11: Normalized Confusion Matrix of the CNNsegment model applied to the $\text{Ortho}_{\text{September}}$

### A1.3 Data pre-processing

To reduce the heterogeneity of crowd-sourced photographs and match them with the UAV perspective, we filtered the photographs based on their acquisition distance and plant leaf visibility. The model achieved an $R^2$= 0.7 and F1 = 0.8 on independent test data for both variables. Using predicted acquisition distance and tree trunk presence information for each photograph, we tested different filtering thresholds and combinations prior to training the $\text{CNN}_{\text{window}}$ model for plant species classification. The best result was achieved by filtering photographs with acquisition distances outside the range of 0.3 to 15 m and excluding photographs that showed tree trunks, with a probability of being a trunk > 0.5.

660 ### A1.4 Citizen science data availability

Table A1: Number of downloaded photographs for selected tree species from the iNaturalist and Pl@ntNet datasets.

| No. | Species | iNaturalist | Pl@ntNet |
|---|---|---|---|
| 1 | Acer pseudoplatanus | 9999 | 3205 |
| 2 | Aesculus hippocastanum | 9998 | 1444 |
| 3 | Betula pendula | 9998 | 1308 |
| 4 | Carpinus betulus | 7165 | 2633 |
| 5 | Fagus sylvatica | 9981 | 3304 |
| 6 | Fraxinus excelsior | 7745 | 3130 |
| 7 | Prunus avium | 9999 | 3022 |
| 8 | Quercus petraea | 1491 | 221 |
| 9 | Sorbus aucuparia | 10000 | 2730 |
| 10 | Tilia platyphyllos | 582 | 1449 |

661 ### A1.5 Segmentation model architecture

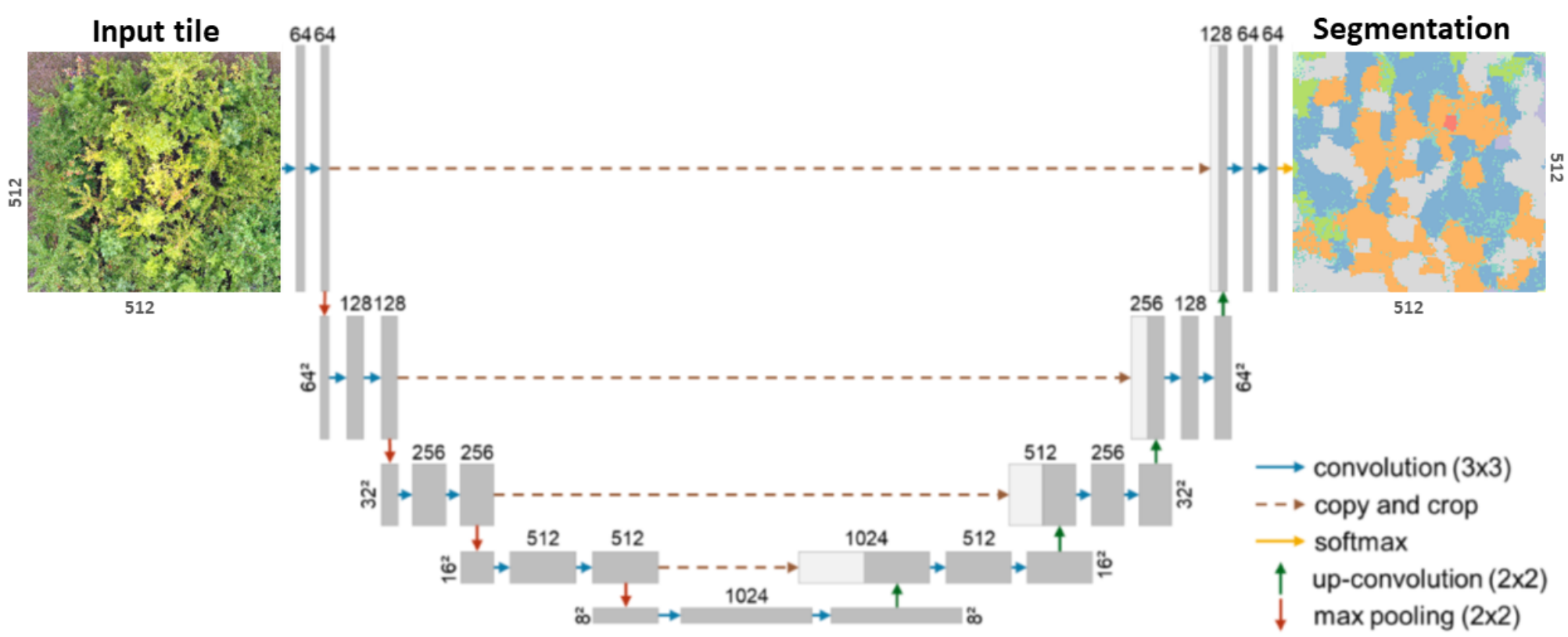

Figure A12: A modified version of the U-Net CNN-architecture for segmenting plant species from UAV orthoimages (Ronneberger et al., 2015).

### A1.6 CNN window species mixture box plot

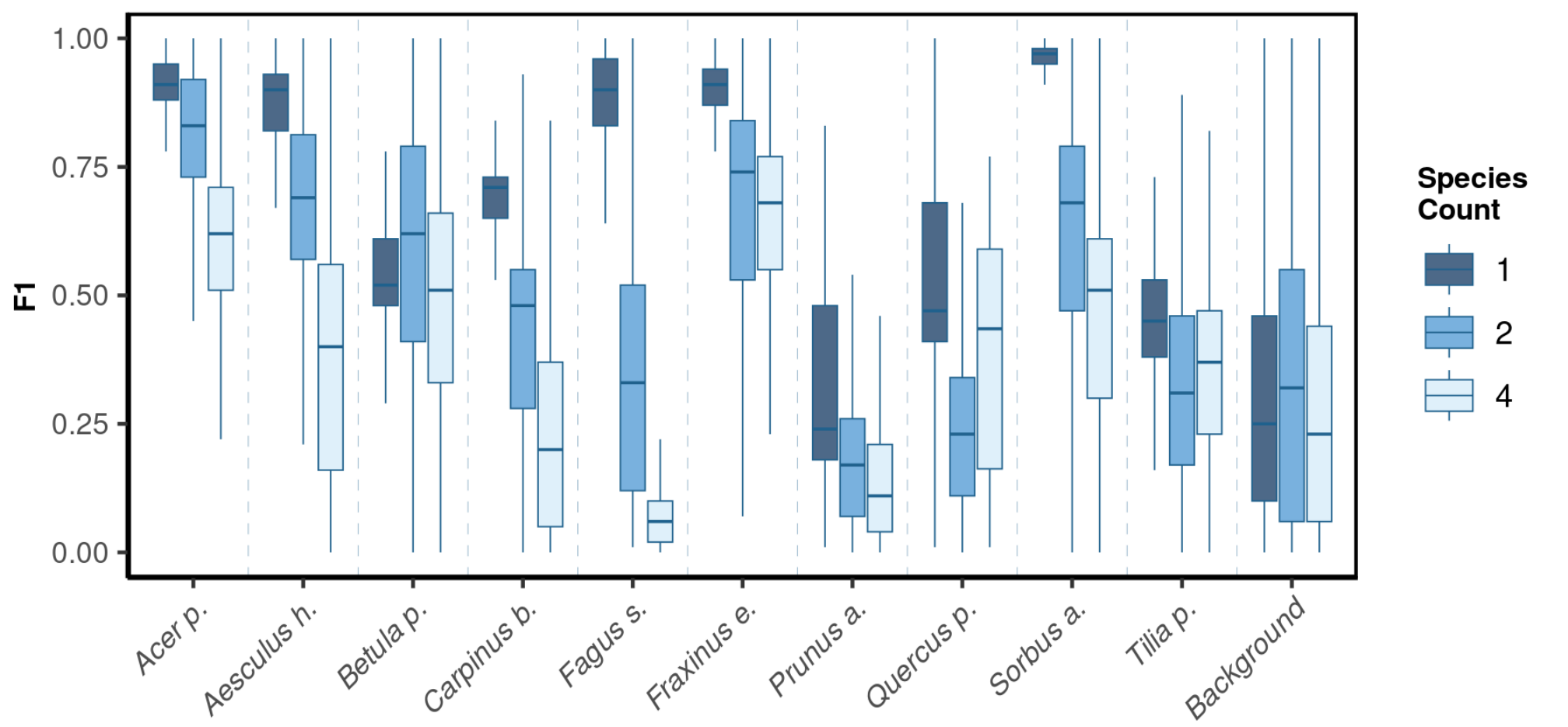

(a) Performance on $\text{Ortho}_{\text{July}}$: The model performance (F1) of the $\text{CNN}_{\text{window}}$ model on Performance on $\text{Ortho}_{\text{July}}$.

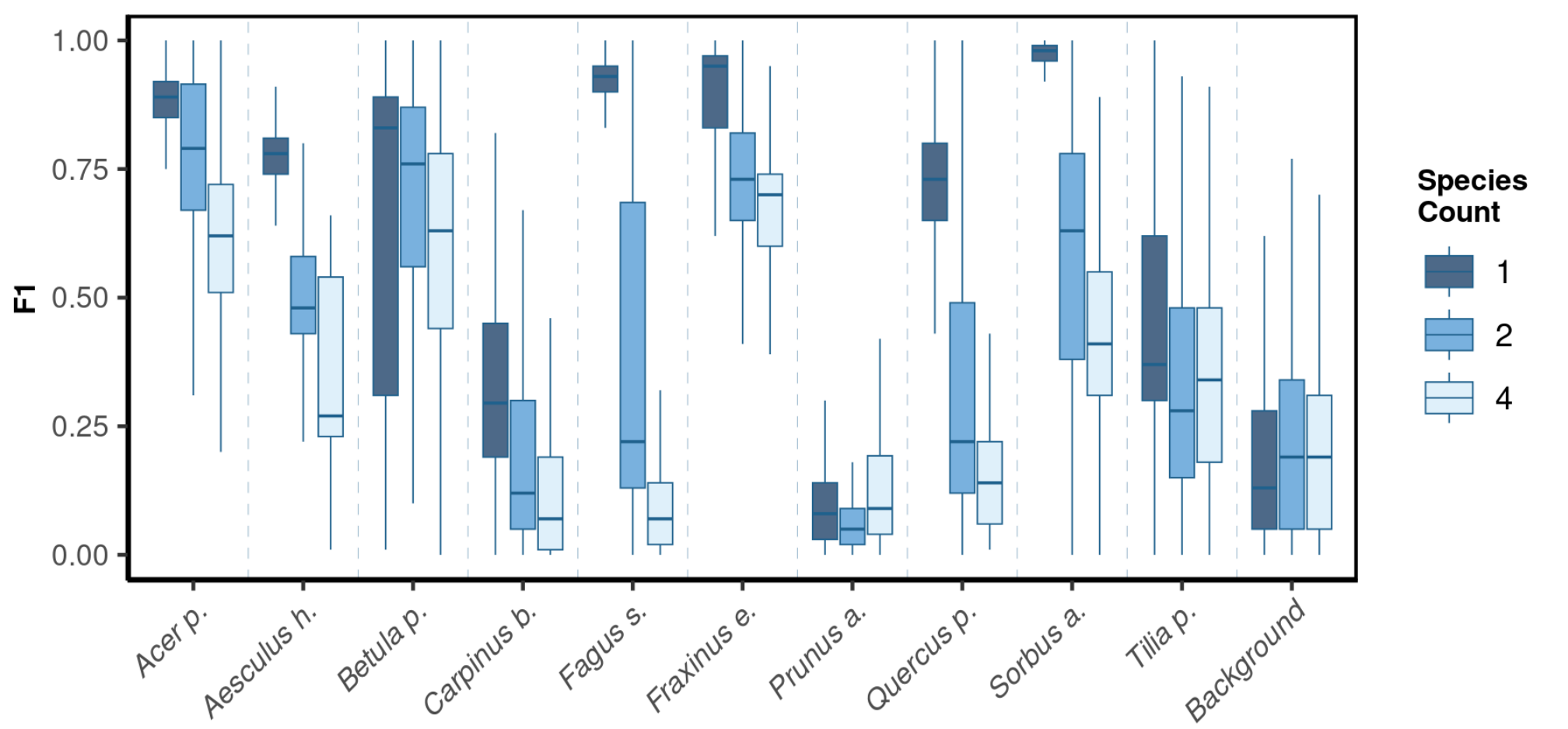

(b) Performance on $\text{Ortho}_{\text{September}}$: The model performance (F1) of the $\text{CNN}_{\text{window}}$ model on Performance on $\text{Ortho}_{\text{July}}$.

Figure A13: The model performance (F1) of the $\text{CNN}_{\text{segment}}$ model across a gradient of canopy complexity in two orthoimages.