# Peer review of "From simple labels to semantic image segmentation: Leveraging citizen science plant photographs for tree species mapping in drone imagery"

_EGUsphere, 2023_

## Author Comment (AC1)

Salim Soltani, Remote Sensing Center for Earth System Research

University of Leipzig, salim.soltani@uni-leipzig.de

To Biogeosciences (BG)

25.03.24

**Ref. No.: egusphere-2023-2576-** *"From simple labels to semantic image segmentation: Leveraging citizen science plant photographs for tree species mapping in drone imagery "*

Dear reviewer,

We would like to thank you for your constructive comments that allowed us to improve the quality of the manuscript and for the time that you spent commenting on the manuscript.

We have addressed the first reviewer comments. We hope that the revised manuscript addresses all the shortcomings of the earlier version.

Kind regards,

Salim Soltani

(on behalf of the Co-authors, Olga Ferlian, Nico Eisenhauer, Hannes Feilhauer , Teja Kattenborn)

| | | **Comments Reviewer #1** | |
|---|---|---|---|
| **ID** | **Line** | **Comment** | **Response** |
| 1 | | The first part of the abstract, that presents an overview of the problem could be shortened to make it more concise (try to summarise each section of the manuscript in 1-3 sentences). | Thank you for pointing out that the abstract is too long. We changed the abstract accordingly: (Line 2-16)

*"Knowledge of plant species distributions is essential for various applications, such as nature conservation, agriculture, and forestry. Remote sensing data, especially high-resolution orthoimages from Unoccupied Aerial Vehicles (UAVs), paired with novel pattern recognition methods, such as Convolutional Neural Networks (CNNs), have been shown to accurately map plant species. Training transferable pattern recognition models for species segmentation across diverse landscapes and data characteristics typically requires extensive training data. Training data are usually derived from labor-intensive field surveys or visual interpretation of remote sensing images, which constrain the training of transferable pattern recognition models. Alternatively, pattern recognition models could be trained on crowd-sourced plant photos and labels from citizen science platforms. That is millions of citizen science-based smartphone photos and the corresponding species label. However, these pairs of citizen science-based photographs and simple species labels (one label for the entire image) cannot be used directly for training state-of-the-art segmentation models used for UAV image analysis, which require per-pixel labels for training (also called masks).*

*Here, we overcome……"* |
| 2 | 250 | l. 250 why did you choose EfficientNetV2L over the other tested backbone architectures? | Thank you for this observation.

We chose EfficientNetV2L over other architectures due to its enhanced classification accuracy in UAV images, outperforming the other tested backbone architectures.

We added an explanation in the sentence: (Line 250-252)

*"After testing different architectures as model backbones, including ResNet-50V2, EfficientNetB07, and EfficientNetV2L, we selected EfficientNetV2L as it resulted in the highest classification accuracies"* |
| 3 | 261 | how much % of the images were assigned NA? Did this influence the model training? | We thank the reviewer for pointing this out. We added this information: (Line 262)

*"......... it was assigned to NA (not available), which accounts for approximately 7.8% of the images."*

Did this influence the model training? overall, yes. A study by Lopatin et al. in 2019 revealed that canopy gaps, serve as a significant source of variability and can lead to misclassification. To address this challenge, we set the pixels belonging to canopy gaps to "NA," effectively excluding them from the model training process. This improved the final classification accuracy. |

| 4 | 240 | Could you explain the term "replacements" (e.g. l. 240)? | We thank the reviewer for pointing out this. We elaborated on the term "replacements" and added the below explanation: (Line 240-242) *"Sampling with replacement randomly duplicates existing photographs for under-represented classes. In this case, classes with fewer than 4,000 photographs."* |
|---|---|---|---|
| 5 | 297-298 | Do you think the amount of misclassified data could be a problem for the training of the segmentation model? (l. 297-298) | Overall, yes. The errors of the moving window-based model can propagate to the segmentation models. This aspect is discussed in section 4.2. (Line 368-385) |
| 6 | | 0.22 cm already seems like very high resolution. Many remote sensing studies focus on making high resolution reference data more usable over large areas (i.e. by adapting it to satellite data). You argue for the use of even finer resolution data in the future. What research objectives could be studies using this very high resolution of UAV data? Is there a research gap for very high prediction accuracy over relatively small areas? Could multispectral/hyperspectral sensors be more useful than higher resolution? | Yes, from the perspective of an average orthoimage for vegetation assessments, the resolution of 0.22 cm may be quite high. However, plant species identification by botanists is usually based on patterns of morphological features (e.g. leaf forms, leaf edges, leaf size, fruits). Thus, if we want to have a similarly high accuracy for species identification, we need to harness high spatial resolution and the 0.22 cm is not particularly high. We believe that spectral resolution will not help in this regard, since most plant species are spectrally very similar (e.g. most plants are green). The most indicative feature of a plant species is its morphology. We added some examples where accurate and spatial detailed information on species distribution is necessary: (Line 35-37) *"Spatially explicit information on plant species is crucial for various applications, including nature conservation, agriculture, and forestry. For instance, species information is required for the identification of threatened or invasive species, the location of weeds or crops in precision farming, or tree species classification for forest inventories."* |
| 7 | 29 | Please remove the "and" between "data" and "by" | Corrected. |
| 8 | 51 | "unleash" might not be the right word; "harness" might be better suited "provided" might be better instead of "given" | We have replaced the two suggested words. *"An effective way to harness the potential of these fine spatial features is provided by deep learning-based pattern-recognition techniques…."* |
| 9 | 56-60 | This sentence is not completely clear to me. Maybe you can reformulate it to make it easier to read. | We simplified the sentence as follows: (Line 56-59) *"Given these high-dimensional computations, efficiently adopting these models to UAV orthoimagery, which often have large spatial extents and high resolution, requires training and applying them sequentially using smaller sub-regions of an orthoimage (e.g., image tiles of 512 by 512 pixels, Fig. 1a). "* |
| 10 | 63 | Please remove "similar", as it is unnecessary | Corrected. |
| 11 | 66 | Consider combining sentence "[…] costly, as training data […]" | We have modified the sentence as suggested:(Line 64-66) |

| | | | "Commonly, the generation of training data is costly, as training data are usually derived from field surveys or visual interpretation of remote sensing images, also known as annotation or labelling." |
|---|---|---|---|
| 12 | 81 | Is the training data limited or just costly/time consuming to generate? | We think that the one is a consequence of the other: Training data is limited because it is costly/time consuming to generate it. |
| 13 | 89 | "platforms" | Corrected. |
| 14 | 90-95 | "mil" or "M"; please remove "of" | Corrected. (Line 89-90)

*"Currently, the iNaturalist project contains over 26 mil globally distributed and annotated photographs of vascular plant species."* |
| 15 | 97 | Please remove "The" before "Pl@ntNet" | Corrected. |
| 16 | 109 | "Ideally, for species mapping applications […]" | Corrected. |
| 17 | 115-120 | This part might fit better in the Methods section | We have removed the technical information from the sentence to better fit the introduction section. (Line 114-117)

*"At first, image classification models were trained with citizen science data and simple labels to predict a species per image. The trained image classification models were then applied sequentially on tiles of UAV-based orthomosaics in a moving-window-like fashion with very high overlap (Fig. 1b)."* |
| 18 | 198 | Please remove "Accordingly" | We have changed it to "Therefore". |
| 19 | 235 | "were afterward rasterized" | Corrected. |
| 20 | 240-241 | What does "sampled with replacement" mean? | Thank you for the observation. We have added a short explanation to improve the clarity of the sentence: (Line 240-242)

*"Sampling with replacement randomly duplicates existing photographs for under-represented classes. In this case, classes with fewer than 4,000 photographs."* |
| 21 | 317 | Please replace "while" with "although", or similar | Corrected. |
| 22 | 337-341 | This might fit better in the Discussion section | We think that it is not trivial to classify this sentence as result or discussion but would like to keep it here to draw a connection to the figure. |
| 23 | 367 | "varying" | Corrected. |
| 24 | 373 | "partially relatively inaccurate" → This is a little vague. Maybe expand upon it a little. | We made this more explicit by changing the sentence: to (Line 374-376)

*"However, although the fact that the segmentation labels created with the CNNwindow approach are partly inaccurate (Fig. 4a, 6), we found that the CNNsegment procedure indeed resulted in higher performance than the CNNwindow procedure."* |
| 25 | 387-389 | Please remove one instance of "plots with more species (two or four)" | Thank you for the observation. We have removed it. |
| 26 | 393 | "higher value" than what? | We have modified it as follows: (Line 395)

*"….in high value for remote sensing applications."* |
| 27 | 442 | Maybe you can find a better phrasing than "diversity of human behaviour" | We have modified as following: (Line 444) |

| | | | *"variability of human behavior"* |
|---|---|---|---|
| 28 | 457 | "often costly" | Modified. |
| 29 | 484 | "large" instead of "excessive" (which means unreasonably much) | Modified. |
| 30 | 485 | "good transferability" | Modified. |
| 31 | Figure 2 | The text font is very small. It would also be better if the labels match the ones used in the text: "OrthoJuly" and "OrthoSeptember" instead of "Ortho 1" and "Ortho 2" | Thank you for the observation. We modified the Figure 2:

- Increased the text font size
- Change the Ortho1 and Ortho2 OrthoJuly and Orthoseptember |
| 32 | Figure 4 | The text font here is also very small. | We have increased the font size from 8pt to 12pt for the Figure labels. |
| 33 | Figure 6 | The height of the transects seems to be different between plots (eg. plot 29 and plot 33). If they are all the same (2 m), please show them with the same extents in the figure as well. | Thank you for the observation. We have harmonized the size of transect plots. |